# 3D Reconstruction with Generalizable Neural Fields using Scene Priors

**Yang Fu**[1][†]    **Shalini De Mello**[2]    **Xueting Li**[2]    **Amey Kulkarni**[2]
**Jan Kautz**[2]    **Xiaolong Wang**[1]    **Sifei Liu**[2]
[1]University of California, San Diego    [2]NVIDIA

## Abstract

High-fidelity 3D scene reconstruction has been substantially advanced by recent progress in neural fields. However, most existing methods train a separate network from scratch for each individual scene. This is not scalable, inefficient, and unable to yield good results given limited views. While learning-based multi-view stereo methods alleviate this issue to some extent, their multi-view setting makes it less flexible to scale up and to broad applications. Instead, we introduce training generalizable Neural Fields incorporating scene Priors (NFPs). The NFP network maps any single-view RGB-D image into signed distance and radiance values. A complete scene can be reconstructed by merging individual frames in the volumetric space WITHOUT a fusion module, which provides better flexibility. The scene priors can be trained on large-scale datasets, allowing for fast adaptation to the reconstruction of a new scene with fewer views. NFP not only demonstrates SOTA scene reconstruction performance and efficiency, but it also supports single-image novel-view synthesis, which is underexplored in neural fields. More qualitative results are available at: https://oasisyang.github.io/neural-prior.

## 1 Introduction

Reconstructing a large indoor scene has been a long-standing problem in computer vision. A common approach is to use the Truncated Signed Distance Function (TSDF) (Zhou et al., 2018; Dai et al., 2017b) with a depth sensor on personal devices. However, the discretized representation with TSDF limits its ability to model fine-grained details, e.g., thin surfaces in the scene. Recently, a continuous representation using neural fields and differentiable volume rendering (Guo et al., 2022; Yu et al., 2022; Azinović et al., 2022; Wang et al., 2022b; Li et al., 2022) has achieved impressive and detailed 3D scene reconstruction.

Although these results are encouraging, all of them require training a distinct network for every scene, leading to extended training durations with the demand of a substantial number of input views.

To tackle these limitations, several works learn a generalizable neural network so that the representation can be shared among different scenes (Wang et al., 2021b; Zhang et al., 2022; Chen et al., 2021; Long et al., 2022; Xu et al., 2022). While these efforts scale up training on large-scale scene datasets, introduce generalizable intermediate scene representation, and significantly cut down inference time, they all rely on intricate fusion networks to handle multi-view input images at each iteration. This adds complexity to the training process and limits flexibility in data preprocessing.

In this paper, we propose to perform 3D reconstruction by learning generalizable **N**eural **F**ields using scene **P**riors (**NFPs**). Such priors are largely built upon depth-map inputs (given posed RGB-D images). By leveraging the priors, our NFPs network allows for a simple and flexible design with single-view inputs during training, and it can efficiently adapt to each novel scene using fewer input views. Specifically, full scene reconstruction is achieved by directly merging the posed multi-view frames and their corresponding fields from NFPs, without the need for learnable fusion blocks.

A direct way to generalize per-scene Nerf optimization is to encode each single-view input image into an intermediate representation in the volumetric space. Yet, co-learning the encoder and the

---

† This work was done while Yang Fu was a research intern at NVIDIA.

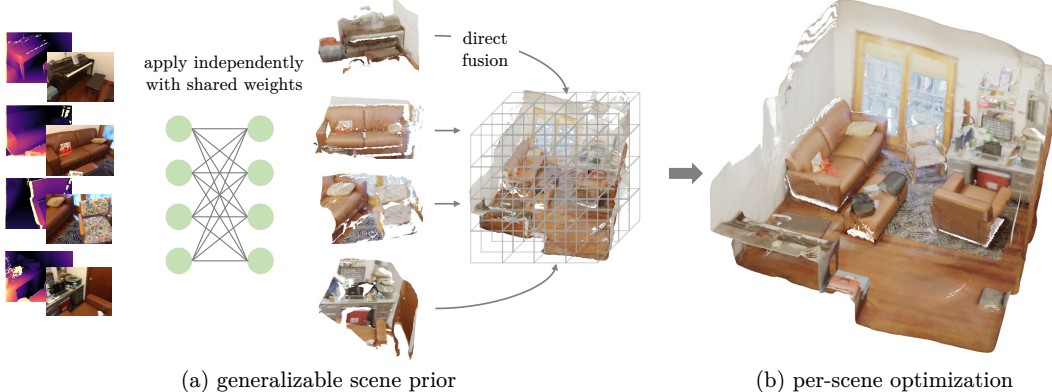

(a) generalizable scene prior           (b) per-scene optimization

Figure 1: We propose Neural Fields scene Prior (NFP) to enable fast reconstruction of geometry and texture of indoor scenes. Our method first (a) learns a generalizable network as a scene prior that obtains a coarse scene reconstruction in a feed-forward manner. Next, we directly fuse the per-view results and (b) perform per-scene optimization in a more accurate and efficient way leading to high-quality surface reconstruction and realistic texture reconstruction.

NeRF presents significant challenges. Given that a single-view image captures only a thin segment of a surface, it becomes considerably harder to discern the geometry compared to understanding the texture. Thus, to train NFPs, we introduce a two-stage paradigm: (i) We train a geometric reconstruction network to map depth images to local SDFs; (ii) We adopt this pre-trained network as a **geometric prior** to support the training of a separate color reconstruction network, as a **texture prior**, in which the radiance function can be easily learned with volumetric rendering (Wang et al., 2021a; Yariv et al., 2021), given the SDF prediction.

Dense voxel grids are a popular choice in many NeRF-based rendering techniques (Yen-Chen et al., 2020; Chen et al., 2021; Liu et al., 2020; Huang et al., 2021; Takikawa et al., 2021; Sun et al., 2022b; Wang et al., 2022b). However, for the single-view input context, they fall short for two main reasons. First, the single-view image inherently captures just a thin and confined segment of surfaces, filling only a minuscule fraction of the entire voxel space. Second, dense voxel grids employ uniform sampling, neglecting surface priors like available depth information. Instead, we resort to a surface representation: we build a set of projected points in the 3D space as keypoint, from where a continuous surface can be decoded. The keypoint representation spans a compact 2D surface representation, allowing dense sampling close to the surface, which significantly enhances scalability.

NFPs can easily facilitate further fine-tuning on large-scale indoor scenes. Given the pretrained geometry and texture network as the scene prior, the single-scene reconstruction can be performed by optimizing the aggregated surface representation and the decoders.

With coarse reconstruction from the generalized network and highly compact surface representation, our approach achieves competitive scene reconstruction and novel view synthesis performance with substantially *fewer views* and *faster convergence speed*. In summary, our contributions include:

- We propose NFPs, a generalizable scene prior that enables fast, large-scale scene reconstruction.
- NFPs facilitate (a) single-view, across-scene input, (b) direct fusion of local frames, and (c) efficient per-scene fine-tuning.
- We introduce a continuous surface representation, taking advantage of the depth input and avoiding redundancy in the uniform sampling of a volume.
- With the limited number of views, we demonstrate competitive performance on both the scene reconstruction and novel view synthesis tasks, with substantially superior efficiency than existing approaches.

## 2 RELATED WORK

Reconstructing and rendering large-scale indoor scenes is crucial for various applications. Depth sensors, on the other hand, are becoming increasingly common in commercial devices, such as Kinect (Zhang, 2012; Smisek et al., 2013), iPhone LiDAR (Nowacki & Woda, 2019), *etc*. Leveraging

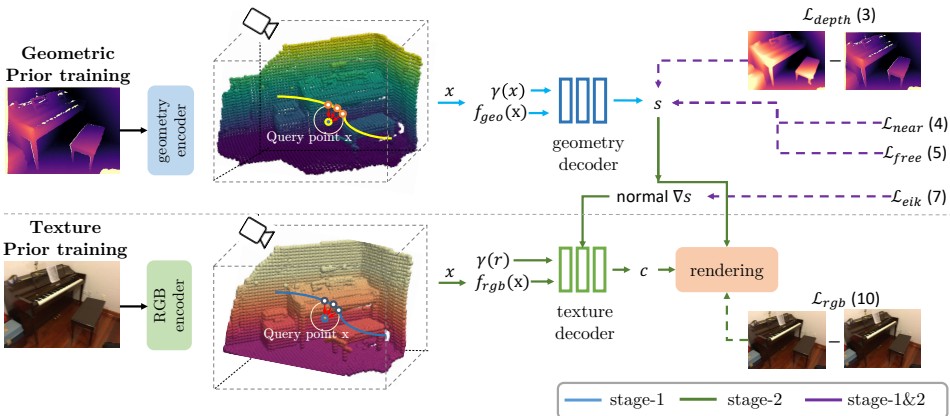

Figure 2: **Overview of NFP**. Given the RGBD input, we first extract the geometric and texture pixel feature using two encoders (Sec. 3.1). Then, we construct the continuous surface representation upon the discrete surface feature (Sec. 3.2). Next, we introduce a two-stage paradigm to learn the generalizable geometric and texture prior, optimized via multiple objectives (Sec. 3.3).

depth information in implicit neural representations is trending. We discuss both these topics in detail, in the following.

**Multi-view scene reconstruction.** Reconstructing 3D scenes from images was dominated by multi-view stereo (MVS) (Schönberger et al., 2016; Schonberger & Frahm, 2016), which often follows the single-view depth estimation (*e.g.*, via feature matching) and depth fusion process (Newcombe et al., 2011; Dai et al., 2017b; Merrell et al., 2007). Recent learning-based MVS methods (Cheng et al., 2020; Düzçeker et al., 2020; Huang et al., 2018; Luo et al., 2019) substantially outperform the conventional pipelines. For instance, Yao et al. (2018); Luo et al. (2019) build the cost-volume based on 2D image features and use 3D CNNs for better depth estimation. Another line of works (Sun et al., 2021; Bi et al., 2017) fuse multi-view depth and reconstruct surface meshes using techniques such as TSDF fusion. Instead of fusing the depth, Wei et al. (2021), Wang et al. (2021b), Zhang et al. (2022), and Xu et al. (2022) directly aggregate multi-view inputs into a radiance field for coherent reconstruction. The multi-view setting enables learning generalizable implicit representation, however, their scalability is constrained as they always require multi-view RGB/RGB-D data during training. Our approach, for the first time, learns generalizable scene priors from single-view images with substantially improved scalability.

**Neural Implicit Scene Representation.** A growing number of approaches (Yariv et al., 2020; Wang et al., 2021a; Yariv et al., 2021; Oechsle et al., 2021; Niemeyer et al., 2020; Sun et al., 2022a) represent a scene by implicit neural representations. Although these methods achieve impressive reconstruction of objects and scenes with small-scale and rich textures, they hardly faithfully reconstruct large-scale scenes due to the shape-radiance ambiguity suggested in (Zhang et al., 2020; Wei et al., 2021). To address this issue, Guo et al. (2022) and Yu et al. (2022) attempt to build the NeRF upon a given geometric prior, *i.e.*, sparse depth maps and pretrained depth estimation networks. However, these methods take a long time to optimize on an individual scene. As mentioned previously, generalizable NeRF representations with mutli-view feature aggregation are studied (Chen et al., 2021; Wang et al., 2021b; Zhang et al., 2022; Johari et al., 2022; Xu et al., 2022). However, they still focus on reconstructing the scene's appearance, *e.g.*, for novel view synthesis, but cannot guarantee high-quality surface reconstruction.

**Depth-supervised reconstruction and rendering.**

With the availability of advanced depth sensors, many approaches seek depth-enhanced supervision of NeRF (Azinović et al., 2022; Li et al., 2022; Zhu et al., 2022; Sucar et al., 2021; Yu et al., 2022; Williams et al., 2022; Xu et al., 2022; Deng et al., 2022) since depth information is more accessible. For instance, Azinović et al. (2022) enables detailed reconstruction of large indoor scenes by comparing the rendered and input RGB-D images. Unlike most methods that use depth as supervision, Xu et al. (2022), Williams et al. (2022) and Dong et al. (2023) build the neural field conditioned on the geometric prior. For example, Point-NeRF pretrains a monocular depth estimation network and generates a point cloud by lifting the depth prediction. Compared to ours, their geometric prior is less integrated into the main reconstruction stream since it is separately learned and detached. Furthermore, these methods only consider performing novel view synthesis (Xu et al., 2022; Deng et al., 2022), where the geometry is not optimized, or perform pure geometric (Yu et al., 2022; Li

et al., 2022; Williams et al., 2022; Azinović et al., 2022) reconstruction. In contrast, our approach makes the scene prior and the per-scene optimization a unified model that enables more faithful and efficient reconstruction for both color and geometry.

# 3    METHOD

Given a sequence of RGB-D images and their corresponding camera poses, our goal is to perform fast and high-quality scene reconstruction. To this end, we learn a generalizable neural scene prior, which encodes an RGB image and its depth map as continuous neural fields in 3D space and decodes them into signed distance and radiance values. As illustrated in Fig. 2, we first extract generalizable surface features from geometry and texture encoders (Sec. 3.1). Then, pixels with depth values are back-projected to the 3D space as keypoints, from which continuous fields can be built with the proposed surface representation (Sec. 3.2). Motivated by previous works (Wang et al., 2021a; Yariv et al., 2021), we utilize two separate MLPs to decode the geometry and texture representations, which are further rendered into RGB and depth values (Sec. 3.3). To obtain high-quality surface reconstruction, we further propose to optimize the neural representation on top of the learned geometric and texture prior for a specific scene (Sec. 3.4).

## 3.1    CONSTRUCTING SURFACE FEATURE

Given an RGB-D image $\{I, D\}$, we first project the depth map into 3D point clouds in the world coordinate system using its camera pose $\{R, t\}$ and intrinsic matrix K. We sub-sample $M$ points via Farthest Point Sampling (FPS), denoted as $\{p_m\}, m \in [0, M-1]$, which are used as keypoints representing the discrete form of surfaces. We extract generalizable point-wise geometry and texture features, as described below, which are further splatted onto these keypoints. Both encoders are updated when training the NFP.

**Geometry encoder.** For each surface point, we apply the K-nearest neighbor (KNN) algorithm to find $K - 1$ points and construct a local region with $K$ points. Thus, we obtain a collection of $M$ local regions, $\{p_m, \{p_k\}_{k \in \Psi_m}\}, \forall m \in [0, M-1]$, where $\Psi_m$ is the neighbor index set of point $p_m$ and $|\Psi_m| = K - 1$. Then, we utilize a stack of PointConv (Wu et al., 2019) layers to extract the geometry feature from each local region $f_m^{geo} = \text{PointConv}(\{p_m, \{p_k\}_{k \in N_m}\})$.

**Texture encoder.** In addition, we extract RGB features for the keypoints via a 2D convolutional neural network. In particular, we feed an RGB image $I$ into an UNet (Ronneberger et al., 2015) with ResNet34 (He et al., 2016) as the backbone, which outputs a dense feature map. Then, we splat the pixel-wise features $f_m^{tex}$ onto the keypoints, according to the projection location of the surface point $p_m$ from the image plane. Thus, each surface point is represented by both a geometry feature and a texture feature, denoted by $f(p_m) = [f_{geo}(p_m), f_{tex}(p_m)]$.

## 3.2    CONTINUOUS SURFACE IMPLICIT REPRESENTATION

Given the lifted keypoints and their projected geometry and texture features, in this section, we introduce how to construct continuous implicit fields conditioned on such discrete representations. We follow a spatial interpolation strategy: for any query point x (e.g., in a typical volume rendering process, it can be a sampled point along any ray), we first find the $K$ nearest surface points $\{p_v\}_{v \in V}$, where $V$ is a set of indices of the neighboring surface points. Then, the query point's feature can be obtained via aggregation of its neighboring surface points. In particular, we apply distance-based spatial interpolation as

$$f(x) = \frac{\sum_{v \in V} \omega_v f(p_v)}{\sum_{v \in V} \omega_v}; \quad \omega_v = \exp\left(-||x - p_v||\right), \tag{1}$$

where $f(x)$ represents either the geometry $f_{geo}(x)$ or the texture $f_{tex}(x)$ feature, and $p_v$ is the position of the $v$-th neighbouring keypoint. With distance-based spatial interpolation, we establish continuous implicit fields for any point from the discrete keypoints.

The continuous representation suffers from two drawbacks: First, when a point is far away from the surface, $f(x)$ is no longer a valid representation, but will still contribute to decoding and rendering.

Second, the distance $\omega_v$ is agnostic to the tangent direction and hence is likely to blur the boundaries. To mitigate the first problem, we incorporate an additional MLP layer that takes into account both the original surface feature $f(\mathrm{p}_v)$ and its relative distance to the query point $\mathrm{x} - \mathrm{p}_v$, and outputs a distance-aware surface feature $f(\mathrm{p}_v^x) = \mathbf{MLP}(f(\mathrm{p}_v), \mathrm{x} - \mathrm{p}_v)$. Subsequently, this refined surface feature $f(\mathrm{p}_v^x)$ replaces the original surface feature in Eq. 1 to obtain the feature of query point x. In addition, we ensure that the sampled points lie near the surface via importance sampling. We resolve the second issue via providing the predicted normal to the decoders as an input. We refer to Sec. 3.3 and 3.4 for details.

## 3.3 GENERALIZABLE NEURAL SCENE PRIOR

To reconstruct both geometry and texture, i.e., a textured mesh, a direct way is to decode the geometry and texture surface representation (Sec. 3.2) into signed distance and radiance values, render them into RGB and depth pixels (Guo et al., 2022; Yu et al., 2022), and then supervise them by the ground-truth RGB-D images.

Unlike the multi-view setting that covers a significant portion of the volumetric space, the single-view input only encompasses a small fraction of it. From our experiments, we found that the joint training approach struggles to generate accurate geometry.

Hence, we first learn a geometric network that maps any depth input to its corresponding SDF (Sec. 3.3.1). Once a coarse surface is established, learning the radiance function initialized by it becomes much easier – we pose it in the second stage where a generalizable texture network is introduced similarly (Sec. 3.3.2).

### 3.3.1 GENERALIZABLE GEOMETRIC PRIOR

We represent scene geometry as a signed distance function, where in our case, it is conditioned on the geometric surface representation $f_{\mathrm{geo}}(x)$ to allow for generalization ability across different scenes. Specifically, along each back-projected ray with camera center $\mathbf{o}$ and ray direction $\mathbf{r}$, we sample $N$ points as $\mathrm{x}_i = \mathbf{o} + d_i\mathbf{r}$, $\forall i \in [0, N-1]$. For each sampled points $\mathrm{x}_i$, its geometry feature $f_{\mathrm{geo}}(\mathrm{x}_i)$ can be computed by equation 1. Then, the geometry decoder $\phi_{\mathrm{G}}$, taking the point position and its geometry feature as inputs, maps each sampled point to a signed distance, which is defined as $\mathbf{s}(\mathrm{x}_i) = \phi_{\mathrm{G}}(f_{\mathrm{geo}}(\mathrm{x}_i), \mathrm{x}_i)$. Note that we also apply positional encoding $\gamma(\cdot)$ to the point position as suggested in Mildenhall et al. (2020). We omit it for brevity.

Following the formulation of NeuS (Wang et al., 2021a), the estimated depth value $\hat{d}$ is the expected values of sampled depth $d_i$ along the ray:

$$\hat{d} = \sum_i^N T_i \alpha_i d_i; \quad T_i = \prod_{j=1}^{i-1}(1 - \alpha_j)$$

$$\alpha_i = \max\left(\frac{\sigma_s(\mathbf{s}(\mathrm{x}_i)) - \sigma_s(\mathbf{s}(\mathrm{x}_{i+1}))}{\sigma_s(\mathbf{s}(\mathrm{x}_i))}, 0\right), \tag{2}$$

where $T_i$ represents the accumulated transmittance at point $\mathrm{x}_i$, $\alpha_i$ is the opacity value and $\sigma_s$ is a Sigmoid function modulated by a learnable parameter $s$.

**Geometry objectives.** To optimize the generalizable geometric representation, we apply a pixel-wise rendering loss on the depth map,

$$\mathcal{L}_{\mathrm{depth}} = |\hat{d} - \mathrm{D}(x, y)|. \tag{3}$$

Inspired by (Azinović et al., 2022; Li et al., 2022), we approximate ground-truth SDF based on the distance to observed depth values along the ray direction, $b(\mathrm{x}_i) = \mathrm{D}(x, y) - d_i$. Thus, for points that fall in the near-surface region ($|b(\mathrm{x}_i)| \leq \tau$, $\tau$ is a truncation threshold), we apply the following approximated SDF loss

$$\mathcal{L}_{\mathrm{near}} = |\mathbf{s}(\mathrm{x}_i) - b(\mathrm{x}_i)| \tag{4}$$

We also adopt a free-space loss (Ortiz et al., 2022) to penalize the negative and large positive predictions.

$$\mathcal{L}_{\mathrm{free}} = \max\left(0, e^{-\epsilon\mathbf{s}(\mathrm{x}_i)} - 1, \mathbf{s}(\mathrm{x}_i) - b(\mathrm{x}_i)\right), \tag{5}$$

where $\epsilon$ is the penalty factor. Then, our approximated SDF loss is

$$\mathcal{L}_{\text{sdf}} = \begin{cases} \mathcal{L}_{\text{near}} & \text{if } b(\mathrm{x}_i) \leq |\tau| \\ \mathcal{L}_{\text{free}} & \text{otherwise} \end{cases} \tag{6}$$

The approximated SDF values provide us with more explicit and direct supervision than the rendering depth loss (Eq. equation 3).

**Surface regularization.** To avoid artifacts and invalid predictions, we further use the Eikonal regularization term (Yariv et al., 2021; Ortiz et al., 2022; Wang et al., 2021a), which aims to encourage valid SDF values via the following,

$$\mathcal{L}_{\text{eik}} = ||\nabla_{\mathrm{x}_i}\mathbf{s}(\mathrm{x}_i) - 1||_2^2, \tag{7}$$

where $\nabla_{\mathrm{x}_i}\mathbf{s}(\mathrm{x}_i)$ is the gradient of predicted SDF w.r.t. the sampled point $\mathrm{x}_i$.

Therefore, we update the geometry encoder and decoder with the generalizable geometry loss as follows,

$$\mathcal{L}_{\text{geo}} = \lambda_{\text{depth}}\mathcal{L}_{\text{depth}} + \lambda_{\text{sdf}}\mathcal{L}_{\text{sdf}} + \lambda_{\text{eik}}\mathcal{L}_{\text{eik}} \tag{8}$$

### 3.3.2 GENERALIZABLE TEXTURE PRIOR

We build the 2nd stage – the generalizable texture network following the pretrained geometry network, as presented in Sec. 3.3.1, which offers the SDF's prediction as an initialization. Specifically, we learn pixel-wise RGB features, as described in Sec. 3.1, and project them onto the corresponding keypoints. Following the spatial interpolation method in Sec. 3.2, we query the texture feature of any sampled point in 3D space. As aforementioned, the spatial interpolation in Eq. equation 1 is not aware of the surface tangent directions. For instance, a point at the intersection of two perpendicular planes will be interpolated with keypoints coming from both planes. Thus, representations at the boundary regions can be blurred. To deal with it, we further concatenate the surface normal $\nabla_{\mathrm{x}_i}\mathbf{s}(\mathrm{x}_i)$ predicted in the first stage with the input to compensate for the missing information.

With a separate texture decoder $\phi_{\text{tex}}$, the color of point $\mathrm{x}_i$ is estimated, conditioned on the texture feature $f_{\text{tex}}(\mathrm{x}_i)$ and the surface normal $\nabla_{\mathrm{x}_i}\mathbf{s}(\mathrm{x}_i)$ ,

$$\mathbf{c}(\mathrm{x}_i) = \phi_{\text{tex}}(f_{\text{tex}}(\mathrm{x}_i), \mathbf{r}, \nabla_{\mathrm{x}_i}\mathbf{s}(\mathrm{x}_i)), \tag{9}$$

where $\mathbf{r}$ is the ray direction. Here we omit the positional encoding of the point's position and ray direction for conciseness. Therefore, the predicted pixel color can be expressed as $\hat{\mathbf{c}} = \sum_i^N T_i\alpha_i\mathbf{c}_i$, where $T_i$ and $\alpha_i$ are defined same as Eq. equation 2. We supervise the network by minimizing the L2 loss between the rendered pixel RGB values and their ground truth values

$$\mathcal{L}_{\text{rgb}} = ||\hat{\mathbf{c}} - \mathrm{I}(x, y)||_2^2. \tag{10}$$

Meanwhile, we jointly learn the geometry network including the PointConv encoder and geometry decoder introduced in Sec. 3.2, via the same $\mathcal{L}_{\text{geo}}$. Thus, the total loss function for generalizable texture representation learning is

$$\begin{aligned} \mathcal{L}_{\text{tex}} = \lambda_{\text{depth}}\mathcal{L}_{\text{depth}} + \lambda_{\text{sdf}}\mathcal{L}_{\text{sdf}} + \lambda_{\text{eik}}\mathcal{L}_{\text{eik}} + \\ \lambda_{\text{rgb}}\mathcal{L}_{\text{rgb}}. \end{aligned} \tag{11}$$

During volumetric rendering, to restrict the sampled points from being concentrated on the surface, we perform importance sampling based on: (i) the predicted surface as presented in Wang et al. (2021a), and (ii) the input depth map. More details are in the supplementary material.

### 3.4 PRIOR-GUIDED PER-SCENE OPTIMIZATION

To facilitate large-scale, high-quality scene reconstruction, we can further finetune the pretrained generalizable geometric and texture prior to individual scenes, with multi-view frames. Specifically, we first directly fuse the geometry and texture feature of multi-view frames via the scene prior networks. No further learnable modules are required, in contrast, to (Chen et al., 2021; Zhang et al., 2022; Li et al., 2022). Then, we design a prior-guided pruning and sampling module, which lets optimization happen near surfaces. In particular, we initialize the grid in the volumetric space via

| Ours-prior 4 min | ManhattanSDF* 640 min | Go-surf 35 min | Ours 15 min | GT |
|---|---|---|---|---|

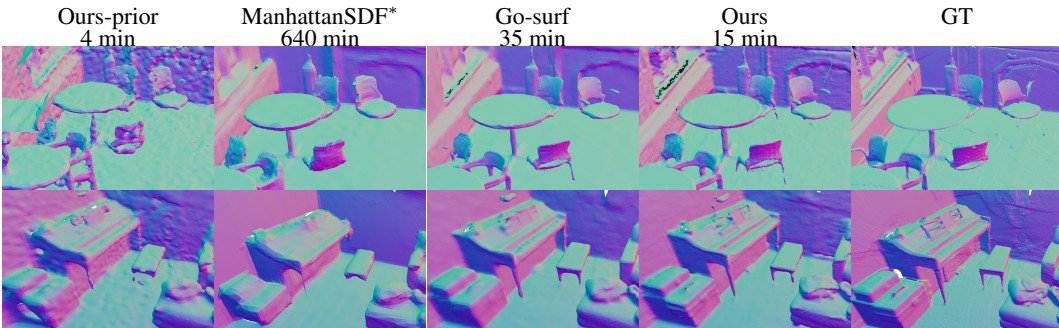

Figure 3: **Qualitative comparisons for mesh reconstruction on ScanNet**. Our method achieves the most complete and fine-detailed reconstruction. The prior reconstruction results and the ground-truth are provided as reference. **Better viewed when zoomed in.**

learn NSP and estimate the SDF value of each grid by its corresponding feature, and remove the grids whose SDF values are larger than a threshold. We note that the generalizable scene prior can be combined with various optimization strategies (Xu et al., 2022; Yu et al., 2022; Wang et al., 2022b). **More details can be found in the supplementary materials.**

During the finetuning, we update the scene-prior feature, and the weights of the MLP decoders to fit the captured images for a specific scene. Besides the objective functions described in Eq. equation 11, we also introduce the smoothness regularization term to minimize the difference between the gradients of nearby points

$$\mathcal{L}_{\text{smooth}} = ||\nabla_{\text{x}_i}\mathbf{s}(\text{x}_i) - \nabla_{\text{x}_i+\sigma}\mathbf{s}(\text{x}_i + \sigma)||_2, \tag{12}$$

where $\sigma$ is a small perturbation value around point $\text{x}_i$. Thus, the total loss function for per-scene optimization is

$$\mathcal{L}_{\text{scene}} = \lambda_{\text{depth}}\mathcal{L}_{\text{depth}} + \lambda_{\text{sdf}}\mathcal{L}_{\text{sdf}} + \lambda_{\text{eik}}\mathcal{L}_{\text{eik}} + \\ \lambda_{\text{rgb}}\mathcal{L}_{\text{rgb}} + \lambda_{\text{smooth}}\mathcal{L}_{\text{smooth}}. \tag{13}$$

## 4 EXPERIMENTS

In this work, we introduce a generalizable network that can be applied to both surface reconstruction and novel view synthesis from RGB-D images in an offline manner. To our best knowledge, there is no prior work that aims for both two tasks. To make fair comparisons, we compare our work with the state-of-the-art (STOA) approaches of each task, respectively.

### 4.1 BASELINES, DATASETS AND METRICS

**Baselines.** To evaluate surface reconstruction, we consider the following two groups of methods: First, we compared our method with RGB-based neural implicit surface reconstruction approaches: ManhattanSDF (Guo et al., 2022) and MonoSDF (Yu et al., 2022) which involve an additional network to extract the geometric prior during training. Second, we consider several RGB-D surface reconstruction approaches that share similar settings with ours: Neural-RGBD (Azinović et al., 2022) and Go-surf (Wang et al., 2022b). In addition, to have a fair comparison, we finetune ManhattanSDF and MonoSDF with ground-truth depth maps as two additional baselines and denoted as ManhattanSDF* and MonoSDF*. We follow the setting in (Guo et al., 2022; Azinović et al., 2022) and evaluate the quality of the mesh reconstruction in different scenes. We note that all the above approaches perform per-scene optimization.

To evaluate the performance in novel view synthesis, we compare our method with the latest NeRF-based methods in novel view synthesis, including NeRF (Mildenhall et al., 2020), NSVF (Liu et al., 2020), NerfingMVS (Wei et al., 2021), IBRNet (Wang et al., 2021b) and NeRFusion Zhang et al. (2022). As most of existing works are only optimized with RGB data, we further evaluate the Go-surf for novel view synthesis from RGB-D images as another baseline. We adopt the evaluation setting in NerfingMVS, where we evaluate our method on 8 scenes, and for each scene, we pick 40 images covering a local region and hold out 1/8 of these as the test set for novel view synthesis.

**Datasets.** We mainly perform experiments on ScanNetV2 (Dai et al., 2017a) for both surface reconstruction and novel view synthesis tasks. Specifically, we first train the generalizable neural scene prior on the ScanNetV2 training set and then evaluate its performance in two testing splits proposed by Guo et al. (2022) and Wei et al. (2021) for surface reconstruction and novel view synthesis, respectively. The GT of ScanNetV2, produced by BundleFusion Dai et al. (2017b), is

| Method | depth | opt. (min) | Acc↓ | Comp↓ | Prec↑ | Recall↑ | F-score↑ |
|---|---|---|---|---|---|---|---|
| ManhattanSDF (Guo et al., 2022) | SfM | 640 | 0.072 | 0.068 | 0.621 | 0.586 | 0.602 |
| MonoSDF (Yu et al., 2022) | network | 720 | 0.039 | 0.044 | 0.775 | 0.722 | 0.747 |
| NeuRIS (Wang et al., 2022a) | network | 480 | 0.051 | 0.048 | 0.720 | 0.674 | 0.696 |
| FastMono (Dong et al., 2023) | network | 30 | 0.042 | 0.056 | 0.751 | 0.678 | 0.710 |
| HelixSurf (Liang et al., 2023) | network | 30 | 0.038 | 0.044 | 0.786 | 0.727 | 0.755 |
| ManhattanSDF* (Guo et al., 2022) | GT. | 640 | **0.027** | 0.032 | 0.915 | 0.883 | 0.907 |
| MonoSDF* (Yu et al., 2022) | GT. | 720 | 0.033 | 0.026 | 0.942 | 0.912 | 0.926 |
| Neural-RGBD (Azinović et al., 2022) | GT. | 240 | 0.055 | 0.022 | 0.932 | 0.918 | 0.925 |
| Go-surf (Wang et al., 2022b) | GT. | 35 | 0.052 | 0.018 | 0.946 | 0.956 | 0.950 |
| Ours-prior (w/o per-scene opt.) | – | – | 0.084 | 0.057 | 0.695 | 0.764 | 0.737 |
| Ours (w per-scene opt.) | GT. | **15** | 0.049 | **0.017** | **0.947** | **0.962** | **0.954** |

Table 1: **Quantitative comparisons for mesh reconstruction on ScanNet.** We compare with a number of baselines. "∗" is our re-implementation with dense ground-truth depth map. "**opt.**" stands for the optimization time for per-scene fine-tuning.

| Method | #frame | Acc ↓ | Comp ↓ | C-$\ell_1$ ↓ | NC ↑ | F-score↑ |
|---|---|---|---|---|---|---|
| BundleFusion (Dai et al., 2017b) | 1,000 | 0.0191 | 0.0581 | 0.0386 | 0.9027 | 0.8439 |
| COLMAP (Schönberger et al., 2016) | 1,000 | 0.0271 | 0.0322 | 0.0296 | 0.9134 | 0.8744 |
| ConvOccNets (Peng et al., 2020) | 1,000 | 0.0498 | 0.0524 | 0.0511 | 0.8607 | 0.6822 |
| SIREN (Sitzmann et al., 2020) | 1,000 | 0.0229 | 0.0412 | 0.0320 | 0.9049 | 0.8515 |
| Neural RGBD (Azinović et al., 2022) | 1,000 | 0.0151 | 0.0197 | 0.0174 | 0.9316 | 0.9635 |
| Go-surf (Wang et al., 2022b) | 1,000 | 0.0158 | 0.0195 | 0.0177 | 0.9317 | 0.9591 |
| Ours | 1,000 | 0.0172 | 0.0192 | 0.0177 | 0.9311 | 0.9529 |
| Go-surf (Wang et al., 2022b) | 30 | 0.0246 | 0.0442 | 0.0336 | 0.9117 | 0.9042 |
| Ours | 30 | **0.0177** | **0.0292** | **0.0234** | **0.9207** | **0.9311** |

Table 2: **Quantitative evaluation of the reconstruction quality on 10 synthetic scenes .** Our method show competitive results when being reconstructed using only 30 frames used per room, in the lower part of the table.

known to be noisy, making accurate evaluations against it challenging. To further validate our method, we also conduct experiments on 10 synthetic scenes proposed by Azinović et al. (2022).

**Evaluation Metrics.** For 3D reconstruction, we evaluate our method in terms of mesh reconstruction quality used in Guo et al. (2022). Meanwhile, we measure the PSNR, SSIM, and LPIPS for novel view synthesis quality.

## 4.2 COMPARISONS WITH THE STATE-OF-THE-ART METHODS

**Surface reconstruction.** Table 1 provides a quantitative comparison of our methods against STOA approaches for surface reconstruction (Guo et al., 2022; Yu et al., 2022; Wang et al., 2022a; Liang et al., 2023). Within our methods, the feed-forward NFPs are denoted as *Ours-prior*, while the per-scene optimized networks are labeled as *Ours*. We list the RGB- and RGB-D-based approaches as in the top and the middle rows, whereas ours are placed in the bottom section. While we include ManhattanSDF (Guo et al., 2022) and MonoSDF (Yu et al., 2022) that are supervised by predicted or sparse depth information as in the top row, to ensure fair comparisons, we re-implement them by replacing the original supervision with ground-truth depth, as in the middle row (denoted by '∗'). Generally, using ground-truth depths can always enhance the reconstruction performance.

**Comparison with NFPs on ScanNet.** In contrast to all the other approaches that all require time-consuming per-scene optimization, the NPFs can extract the geometry structure through a single forward pass. The results in Table 1 demonstrate that, even without per-scene optimization, the NFPs network not only achieves performance on par with RGB-based approaches but also operates hundreds of times faster. Note in contrast to all the other approaches in Table 1 that use around 400 frames to optimize the scene-specific neural fields, *Ours-prior* only takes around 40 frames per scene as inputs to achieve comparable mesh reconstruction results without per-scene optimization.

**Comparison with optimized NFPs on ScanNet.** We further perform per-scene optimization on top of the NFPs network. Compared with methods using additional supervision or ground truth depth maps, our method demonstrates more accurate results on the majority of the metrics. More importantly, our method is either much faster, compared with the SOTA approaches. Some qualitative results are shown in Fig. 3 and more results can be found in the supplementary materials.

**Comparison on synthetic scenes.** Table 2 compares our approach with the most recent works on neural surface reconstruction from RGB-D images. The results demonstrate that our method achieves

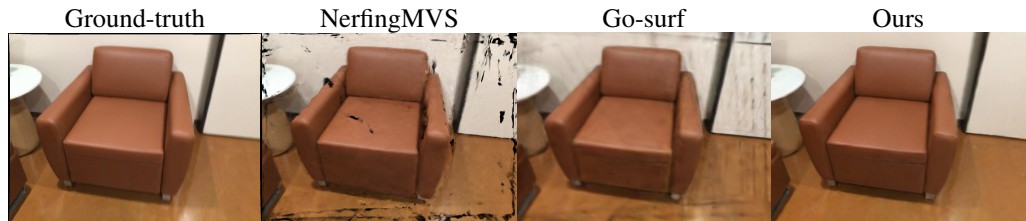

| Ground-truth | NerfingMVS | Go-surf | Ours |

Figure 4: **Qualitative comparison for novel view synthesis on ScanNet.** We compare our method with baselines which achieves the competitive geometry reconstruction performance. Our approach produces more realistic rendering results than two other baselines.

comparable performance with most existing works, even when optimizing with a limited number of frames, such as 1,000 vs 30.

**Results on novel view synthesis.** To validate the learned radiance representation, we further conduct experiments on novel view synthesis. The quantitative results and qualitative results are shown in Table 3 and Fig. 4. Table 3 shows that the proposed method achieves comparable if not better results compared to SOTA novel view synthesis methods (Wang et al., 2021b; Zhang et al., 2022; Liu et al., 2020). We note that our method outperforms Go-surf in this instance, even when both methods achieve comparable geometric reconstruction performance. This suggests that our learned prior representation offers distinct advantages for novel view synthesis. In addition, from Fig. 4, both NerfingMVS (Wei et al., 2021) and Go-surf (Wang et al., 2022b) fail on scenes with complex geometry and large camera motion. The generalized representation enables the volumetric rendering to focus on more informative regions during optimization and improves its performance for rendering RGB images of novel views.

| Method | PSNR↑ | SSIM↑ | LPIPS↓ |
|---|---|---|---|
| NeRF (Mildenhall et al., 2020) | 24.04 | 0.860 | 0.334 |
| NSVF (Liu et al., 2020) | 26.01 | 0.881 | – |
| NeRFingMVS (Wei et al., 2021) | 26.37 | 0.903 | 0.245 |
| IBRNet (Wang et al., 2021b) | 25.14 | 0.871 | 0.266 |
| NeRFusion (Zhang et al., 2022) | 26.49 | **0.915** | **0.209** |
| Go-surf (Wang et al., 2022b) | 25.47 | 0.894 | 0.420 |
| Ours | **26.88** | 0.909 | 0.244 |

Table 3: **Quantitative comparisons for novel view synthesis on ScanNet.** The best two results of different metrics are highlighted.

### 4.3 ABLATION STUDIES

We further perform the ablation studies to evaluate the effectiveness and the efficiency of the neural prior network. **Effectiveness of generalized representation.** Table 4 shows the results with and without the generalized representation. For the model without generalized representation, we randomly initialize the parameters of feature grids and decoders while keeping the other components unchanged. We observe that the model integrated with geometry prior and/or color prior can consistently improve the performance on 3D reconstruction and novel view synthesis.

| Geo. prior | Acc↓ | Comp↓ | F-score↑ |
|---|---|---|---|
| | 0.079 | 0.031 | 0.851 |
| ✓ | **0.046** | **0.030** | **0.862** |

| Color prior | PSNR↑ | SSIM↑ | LPIPS↓ |
|---|---|---|---|
| | 25.87 | 0.899 | 0.415 |
| ✓ | **26.88** | **0.909** | **0.246** |

Table 4: **Ablation studies on geometric and texture prior.** We report both mesh reconstruction metrics and novel view synthesis metrics.

**Fast optimization.** Our approach can achieve high-quality reconstruction at approximately 1.5K iterations within 15 minutes. As illustrated in Fig. 5, our method achieves a high F-score at the very early training stage, while Manhattan SDF* (Guo et al., 2022) and MonoSDF* (Yu et al., 2022) take much more iterations to reach a similar performance.

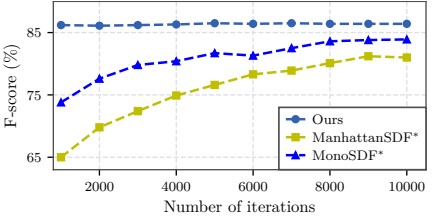

Figure 5: **Ablation studies on the number of training iterations for per-scene optimization.**

## 5 CONCLUSION

In this work, we present a generalizable scene prior that enables fast, large-scale scene reconstruction of geometry and texture. Our model follows a single-view RGB-D input setting and allows non-learnable direct fusion of images. We design a two-stage paradigm to learn the generalizable geometric and texture networks. Large-scale, high-fidelity scene reconstruction can be obtained with efficient fine-tuning on the pretrained scene priors, even with limited views. We demonstrate that our approach can achieve state-of-the-art quality of indoor scene reconstruction with fine geometric details and realistic texture.

**Acknowledgement** This work was supported, in part, by NSF CAREER Award IIS-2240014, the Qualcomm Innovation Fellowship, Amazon Research Award.

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

# A APPENDIX

In the supplementary document, we introduce more implementation details (Sec. A), a comparison with RGB-D surface reconstruction on ScanNet.(Sec. B.1, Sec. B.2 and Sec. B.3), single-view novel view synthesis (Sec. B.4) and qualitative results (Sec. B.5 and Sec. B.6). We specifically discussed the importance of sampling methods w.r.t the two-stage generalizable prior training, which plays an important role in the surface representation, as described in Sec. 3.2 in the main paper. We provide additional experiments including (i) quantitative results of the neural scene prior, *i.e.*, without per-scene optimization, (ii) quantitative comparisons with state-of-the-art RGB-D surface reconstruction approaches, (iii) quantitative comparisons to the MVS-based approaches, (iv) single-view novel view synthesis, and (v) qualitative results on ScanNet and self-collected data. **Videos of full-size room reconstruction are included and recommended to watch.**

# A IMPLEMENTATION DETAILS

## A.1 GENERALIZABLE NEURAL SCENE PRIOR

The generalizable neural scene prior is trained on the training split of ScanNet Dai et al. (2017a). We discuss some details of every component including the geometry encoder, texture encoder, generalizable geometric prior module, and generalizable texture prior in this section.

**Geometry and Texture Encoder.** For the geometry encoder, we first sample 512 keypoints from all the points projected from 2D pixels, via Farthest Point Sampling (FPS) for each frame. For each surface point, we apply the K-Nearest-Neighbor algorithm to select 16 adjacent points. Then, we adopt two PointConv Wu et al. (2019) layers, to extract the geometry feature whose output channels are set to 64. To extract the RGB feature we use a U-Net Ronneberger et al. (2015) with ResNet34 He et al. (2016) as the backbone network. We further use an additional convolutional layer to output a per-point feature with the dimension as 32. All the encoder modules are jointly trained with the whole pipeline.

**Generalizable Geometric Prior.** Given an RGB-D image and its corresponding camera pose, we first randomly sample 256 rays from regions where depth values are valid, *e.g.*, non-zero. Then for each ray, we define a small truncation region near the ground-truth depth where 32 points are sampled uniformly. We then use two MLPs to map the geometry features to SDF values. The hyperparameters $\lambda_{\text{depth}}$, $\lambda_{\text{sdf}}$ and $\lambda_{\text{eik}}$ are set to $1.0, 1.0$ and $0.5$, respectively.

**Generalizable Texture Prior.** Initialized with the geometric prior, we learn the texture prior via the volumetric rendering loss Wang et al. (2021a); Yariv et al. (2021). Different from the sampling strategy used in geometric prior learning, we restrict the importance sampling to the samples concentrated on the surface as described in Sec.3.4 of our main paper. In particular, we first sample 2048 rays from each RGB-D image where we uniformly sample 64 points in the predefined near-far region. Following Wang et al. (2021a), then, we sample 48 points that are close to the predicted surface. This sampling strategy is employed during both training and inference. Additionally, during training, for rays with non-zero depth values, we further sample 16 points within the truncation region around the ray's depth. Therefore, 128 points are sampled along each ray. For each point, we utilize 2 MLPs in the texture decoder to estimate its RGB value. The hyperparamters $\lambda_{\text{depth}}$, $\lambda_{\text{sdf}}$, $\lambda_{\text{eik}}$ and $\lambda_{\text{rgb}}$ are set to $1.0, 1.0, 0.5$ and $10.0$, respectively.

**Scene prior extraction and fusion.** To leverage multiple views of RGB-D frames, with the scene prior networks, we can directly aggregate the keypoints along with their geometry and texture features from these frames in the volumetric space. Then, the colored surface reconstruction can be decoded from the fused representation following the same procedure in Sec.3.1 and 3.2. No further learnable modules are required, in contrast, to Chen et al. (2021); Zhang et al. (2022); Li et al. (2022).

**Prior-guided pruning and sampling.** To optimize a single scene, we discard the encoders and treat the volume feature representation as learnable to be optimized together with the decoders. To further speed up the optimization, we accelerate the feature query process of sampled points, i.e., instead of optimizing the unstructured keypoints, from which the feature extraction can be inefficient, we introduce the prior-guided voxel pruning to leverage the advantage of voxel-grid sampling and surface representation. Specifically, we initialize uniform grids in the volumetric space and then query

each grid feature. Instead of optimizing a large number of uniform grids, we remove the redundant grids adaptively based on the geometric prior using the Algo. 1 described below. To concentrate the sampled ray points near the surface, we apply an importance sampling strategy, similar to that used in training the generalizable texture prior, to mask out those far away from the surface. Starting from a large threshold at the early training stage, we decrease it gradually with more training iterations to prune more unnecessary grids. A similar procedure is also applied to the coarsely sampled points to remove some useless points and help more points concentrate around the surface region. Notably, compared to the voxel-based approach Wang et al. (2022b) having more than $4,000,000$ uniform grids to be optimized, the number of learnable keypoints in our case is around $40,000$ – a 100x reduction in computational complexity.

---

**Algorithm 1:** Prior-guided voxel pruning

**Input:** Grid feature $\{f_i\}_{i=1:N}$;
Grid position $\{x_i\}_{i=1:N}$;
Positional encoding $\gamma(\cdot)$;
Geometry decoder $s(\cdot)$;
Number of grids $N$;
Number of iterations $T$
**Output:** Grid feature after prune $\{f_j\}_{i=1:M}$
**Initialization:** $\tau_0 = 0.16$
**for** $t = 1: T$ **do**
    $\tau = \max(0.005, \ 0.8^{\frac{20t}{T}} \cdot \tau_0)$
    **for** $i = 1 : N$ **do**
        $s_i \leftarrow s(f_i, \gamma(x_i))$;
        **if** $|s_i| \geq \tau$ **then**
            Prune $i$-th grid
        **end**
    **end**
**end**

---

# B ADDITIONAL EXPERIMENTS

## B.1 COMPARISON WITH RGB-D SURFACE RECONSTRUCTION ON SCANNET

**Computational Resource.** The geometric and texture priors network are trained on 8 NVIDIA V100 GPUs for 2 days until convergence. The per-scene optimization step is trained and tested on a single NVIDIA V100 GPU. All baselines reported in our paper are tested using the same computational resources.

**Clarification of Table 2 in the main paper.** Table 2 of the main paper presents the comparison of our method with ManhattanSDF (Guo et al., 2022) and MonoSDF (Yu et al., 2022) with the depth supervision. For fairness, we keep every component of each method by involving an additional depth loss. Unlike some RGB-D surface reconstruction methods, we did not optimize the camera pose while during training. In the course of these modifications, both ManhattanSDF and MonoSDF were observed to have an architecture quite similar to NeuralRGBD Azinović et al. (2022). Given these circumstances, we are confident that comparing our approach with ManhattanSDF and MonoSDF on ScanNet is indeed fair and effective.

**Comparison with Go-surf (Wang et al., 2022b) and NeuralRGBD (Azinović et al., 2022).** We compare our method with Go-surf and Neural RGB-D in Table 5. To have a fair comparison, instead of optimizing camera poses and neural scene representation jointly, we fix the original camera poses as provided by ScanNet Dai et al. (2017a). Follow the same setting in the main paper, we report the performance of different models training with dense and sparse training views. As shown in Table 5, our approach achieves better performance over all metrics. More importantly, although Go-surf achieves similar performance within relatively similar time, it cannot produce any reasonable results without optimization as demonstrated by our approach.

| Method | # frames | opt. time | Prec↑ | Recall↑ | F-score↑ |
|---|---|---|---|---|---|
| Neural-RGBD Azinović et al. (2022) | 400 | 240 | 0.932 | 0.918 | 0.925 |
| Go-surf Wang et al. (2022b) | 400 | 35 | 0.946 | 0.956 | 0.950 |
| Ours | 400 | **15** | **0.947** | **0.962** | **0.954** |
| Neural-RGBD Azinović et al. (2022) | 40 | 240 | 0.837 | 0.855 | 0.846 |
| Go-surf Wang et al. (2022b) | 40 | 35 | 0.842 | 0.861 | 0.851 |
| Ours | 40 | **15** | **0.858** | **0.866** | **0.862** |

Table 5: **Quantitative comparisons for mesh reconstruction on ScanNet.**

| Method | per-scene optim | opt. (min) | Acc↓ | Comp↓ | Prec↑ | Recall↑ | F-score↑ |
|---|---|---|---|---|---|---|---|
| Manhattan SDF (Guo et al., 2022) | ✓ | 640 | 0.072 | 0.068 | 0.621 | 0.586 | 0.602 |
| MonoSDF (Yu et al., 2022) | ✓ | 720 | **0.039** | **0.044** | **0.775** | 0.722 | **0.747** |
| Ours-prior | ✗ | ≤ 5 | 0.084 | 0.057 | 0.695 | **0.764** | 0.737 |

Table 6: **Quantitative comparisons of neural scene prior on ScanNet.** Both Manhanttan SDF and MonoSDF require to optimize on a specific scene for several hours, while the proposed neural scene prior can achieve comparable performance without any optimizatin.

## B.2 MODEL EFFICIENCY

We take Go-surf Wang et al. (2022b), which is so far one of the most efficient offline scene reconstruction approach, as the reference. Compared to it achieving an average run-time of 35 mins per scene, our Neural Scene Prior network takes only **5 mins** (note that the Neural Scene Prior is a feed-forward network). The full pipeline leveraging the per-scene optimization stage takes an average run-time of 15 minutes, which is still obviously more efficient. More importantly, our model takes a surface representation that facilitates scaling up to larger scenes, compared to dense voxels used in Go-surf. A comprehensive comparison of running time can be found in Table 1 of the main paper.

## B.3 COMPARISON WITH MVS-BASED METHODS

We show quantitative comparisons of our method with the state-of-the-art approaches on surface reconstruction in Table 7. Different from what Table 1 reported in the main paper, we mainly compare with the MVS-based methods here. For a fair comparison, we follow the evaluation script used in Zou et al. (2022) for computing 3D metrics on the ScanNet testing set. The top part of Table 7 includes offline methods while the middle one contains online methods with the fusion strategy. The bottom part of the table shows the methods that are finetined on individual scenes. Compared to most MVS-based works that use a fusion strategy, our method achieves much better results in terms of F-score and normal consistency. Moreover, our method outperforms MonoNeuralFusion Zou et al. (2022), which also performs finetuning for individual scenes, by a large margin.

## B.4 NOVEL VIEW SYNTHESIS

**Novel View Synthesis.** We show more qualitative results on novel view synthesis on ScanNet Dai et al. (2017a) in Fig. 6 following the same setting described in the main paper. Both NerfingMVS (Wei et al., 2021) and Go-surf (Wang et al., 2022b) fail on scenes with complex geometry and large camera motion (bottom two rows). The generalized representation enables the volumetric rendering to focus on more informative regions during optimization and improves its performance for rendering RGB images of novel views.

**Single-view Novel View Synthesis.** We demonstrate that NFP enables high-quality novel view synthesis from single-view input (Fig. 7, mid), which has been rarely explored especially at the scene level, and potentially enables interesting applications, *e.g.*, on mobile devices.

## B.5 QUALITATIVE RESULTS OF MESH RECONSTRUCTION

We show qualitative comparisons of our method with other baselines in Fig. 8. It demonstrates that the reconstructed mesh results of our approach are consistently more coherent and detailed than others. In addition, we show the qualitative results of textured mesh for different scenes that obtained via neural scene prior in Fig. 9. **More video demos of texture mesh reconstruction can be found in the supplementary video.**

Ground-truth          NerfingMVS          Go-surf          Ours

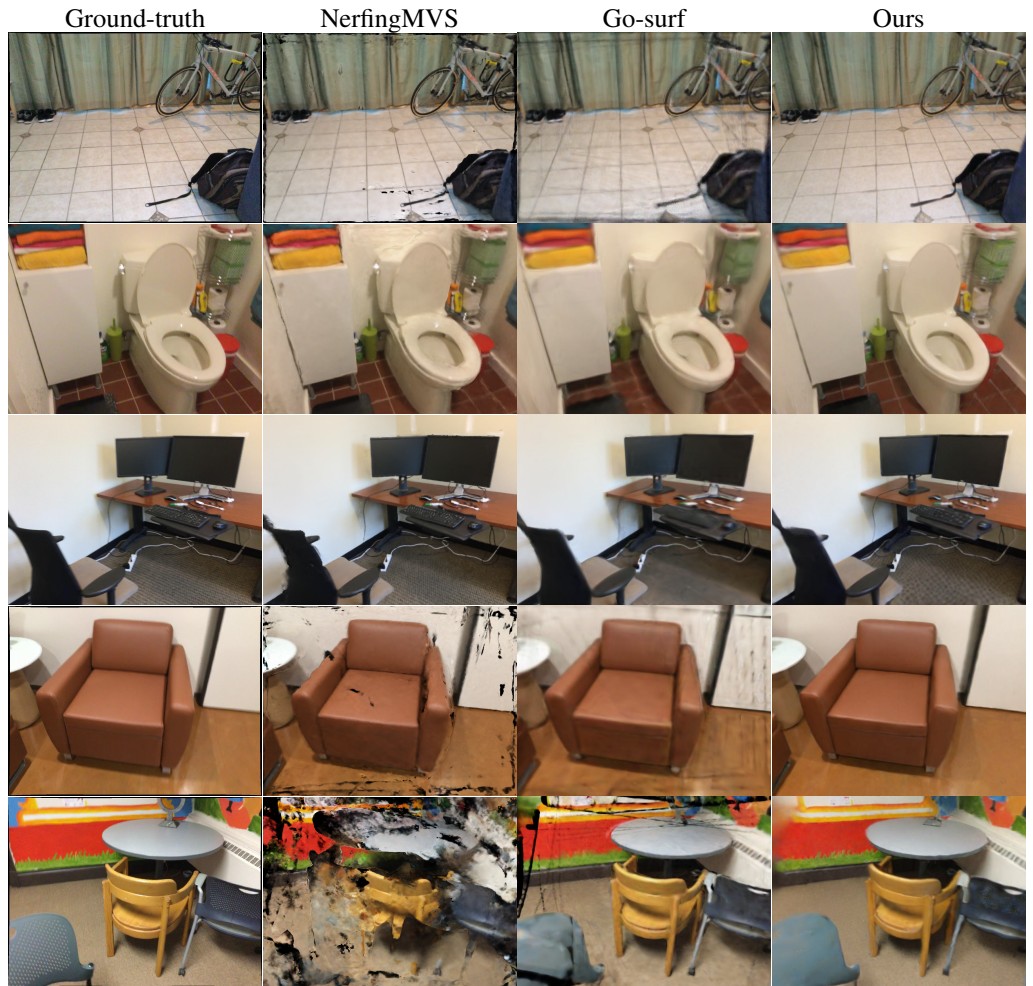

Figure 6: **Qualitative comparison for novel view synthesis on ScanNet.**

Source view          Novel View          Ground-truth

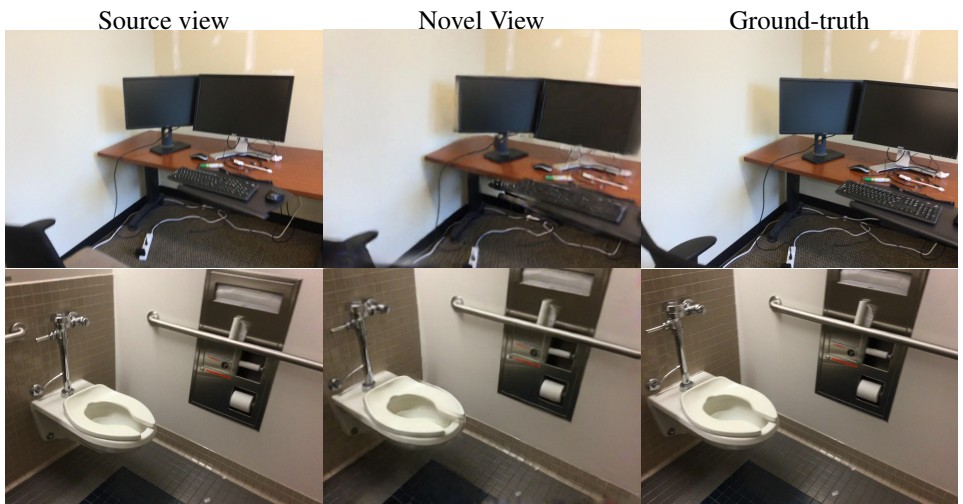

Figure 7: **Qualitative results for single-view novel view synthesis.** The left column shows the training source view, and the appearance reconstruction of the novel view are reported in the second column. The ground-truth images are listed at the last column as reference. **Better viewed when zoomed in.**

|  | Acc ↓ | Comp ↓ | Chamfer ↓ | Precision ↑ | Recall ↑ | F-score ↑ | NC ↑ |
|---|---|---|---|---|---|---|---|
| FastMVSNet Yu & Gao (2020) | 0.052 | 0.103 | 0.077 | 0.652 | 0.538 | 0.588 | 0.701 |
| PointMVSNet Chen et al. (2019) | 0.048 | 0.115 | 0.082 | 0.677 | 0.536 | 0.595 | 0.695 |
| Atlas Murez et al. (2020) | 0.072 | 0.078 | 0.075 | 0.675 | 0.609 | 0.638 | 0.819 |
| GPMVS Hou et al. (2019) | 0.058 | 0.078 | 0.068 | 0.621 | 0.543 | 0.578 | 0.715 |
| DeepVideoMVS Duzceker et al. (2021) | 0.066 | 0.082 | 0.074 | 0.590 | 0.535 | 0.560 | 0.765 |
| TransformerFusion Azinović et al. (2022) | 0.055 | 0.083 | 0.069 | 0.728 | 0.600 | 0.655 | - |
| NeuralRecon Sun et al. (2021) | **0.038** | 0.123 | 0.080 | 0.769 | 0.506 | 0.608 | 0.816 |
| MonoNeuralFusion Zou et al. (2022) | 0.039 | 0.094 | 0.067 | 0.775 | 0.604 | 0.677 | 0.842 |
| Ours | 0.086 | **0.068** | **0.077** | **0.917** | **0.889** | **0.875** | **0.878** |

Table 7: **Quantitative comparisons of mesh reconstruction on ScanNet.**

### B.6 MESH RECONSTRUCTION ON THE LARGE-SCALE SCENE

Our results demonstrate that the neural scene prior we propose can generalize well to large-scale scenes, as shown in Fig 10. In contrast to the previous four scenes, we selected a larger room from ScanNet Dai et al. (2017a) and applied our pre-trained model directly. The left figure in Fig 10 displays the mesh reconstruction obtained from the neural scene prior. Remarkably, our approach successfully recovers the geometry structure of the entire room with very sparse views (60 frames), without requiring any optimization process. Furthermore, by optimizing the prior on this scene for only 20 minutes on a single NVIDIA V100 GPU, we were able to achieve high-quality mesh reconstruction.

### B.7 MESH RECONSTRUCTION ON THE SELF-CAPTURED SCENE

To further demonstrate the robustness of the neural scene prior, we evaluate the pretrained model on a self-captured living room, and the reconstructed mesh w./w.o texture is shown in Fig. 11. Impressively, even without per-scene optimization, the proposed neural scene prior is capable of feasibly reconstructing a textured mesh.

## C LIMITATION

The proposed neural scene prior could extract the geometric and texture prior for arbitrary scenes, but it does require the sparse RGB-D images as the input. To adapt this neural scene prior for RGB images, one possibility would be to initially create a sparse point cloud using Structure from Motion (SfM) on RGB images. However, as of our submission time, we haven't yet experimented with this particular setup. Exploring this pathway in future research could certainly yield intriguing findings.

## D REPRODUCIBILITY STATEMENT

All experiments in this paper are reproducible. We are committed to releasing the source codes once accepted.

## E USE OF EXISTING ASSETS.

As mentioned in the NeurIPS 2023 checklist, we describe the existing assets we used in our paper and the corresponding license of these assets.

## F PERSONAL DATA AND HUMAN SUBJECTS

The dataset does not include the facial or other identifiable information of humans.

## G ETHICAL CONCERNS.

The datasets used are standard benchmarks proposed in previous works. Despite applying supervised learning, there may still be potential bias in our model trained with these datasets.

Manhattan SDF*      MonoSDF*        Ours            GT

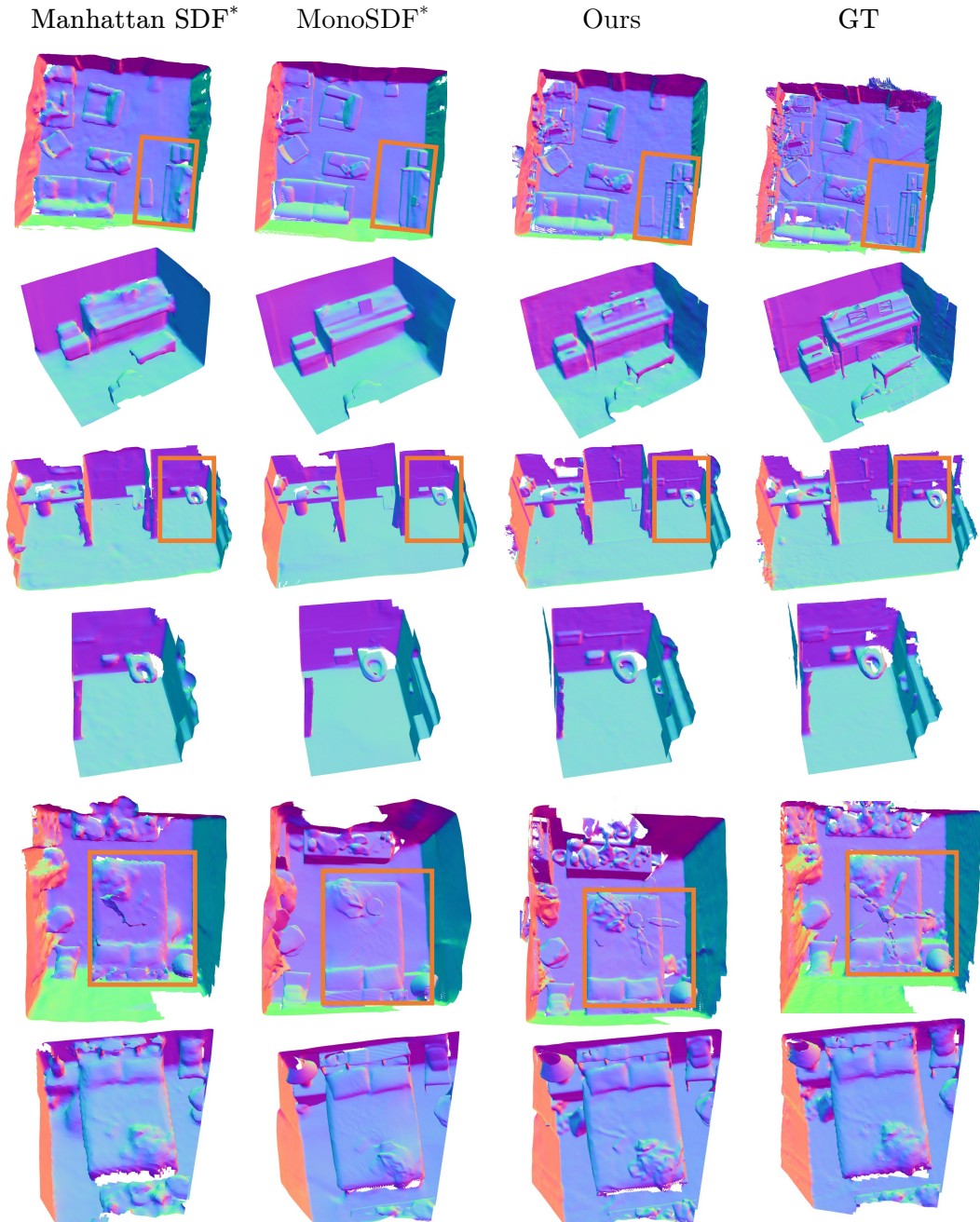

Figure 8: **Qualitative comparisons of mesh reconstruction on ScanNet**. Selected local regions are highlighted by the orange bounding box. **Better viewed when zoomed in.**

**Datasets** Most of the experiments are conducted on ScanNet dataset and 10 synthetic scenes collected by Dai et al. (2017a) and Azinović et al. (2022) which are released on their official website and public to everyone for non-commercial use.

**Code.** Our code is built upon the Pytorch Paszke et al. (2019). And we leverage the code from the released codes by nerfstudio Tancik et al. (2023) under the Apache License.

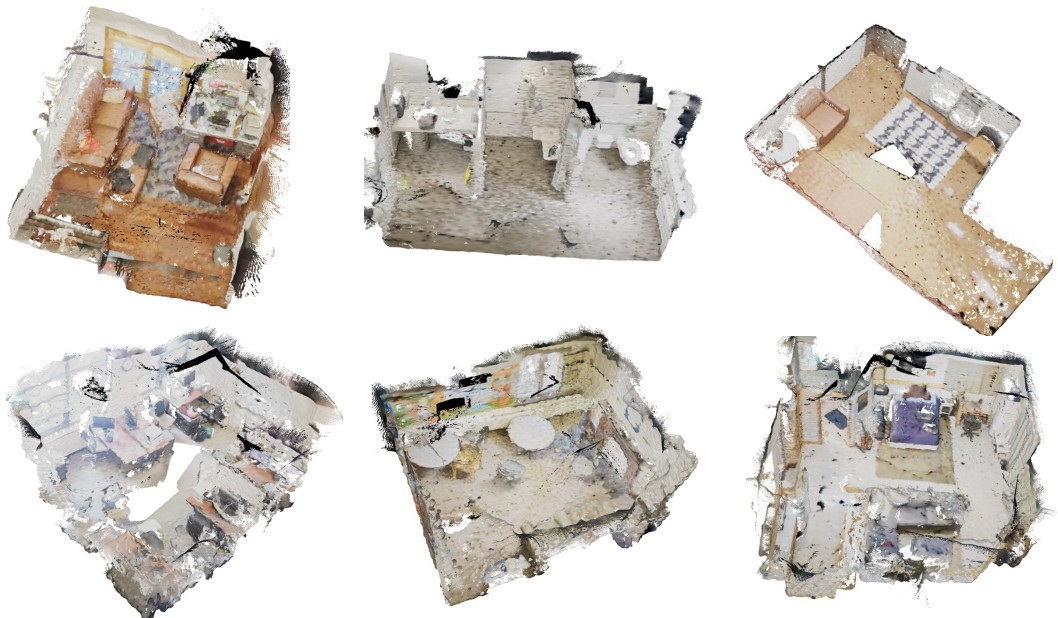

Figure 9: **Qualitative results of Neural Scene Prior on ScanNet**.

Scene Prior                                           Per-scene Optimization

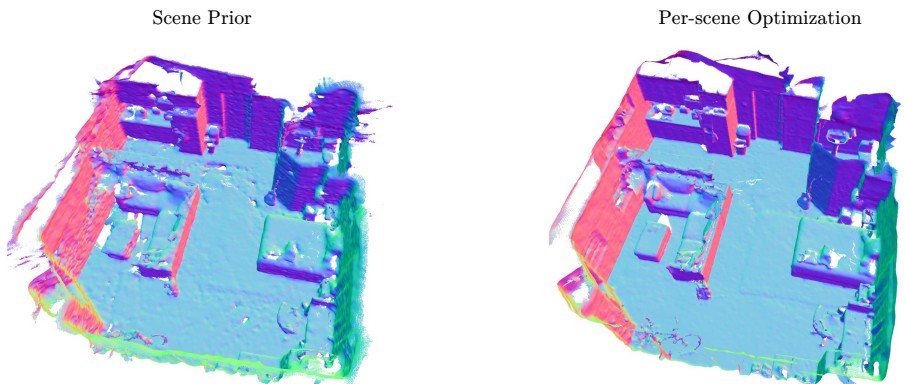

Figure 10: **Mesh reconstruction results on the large-scale scene.**

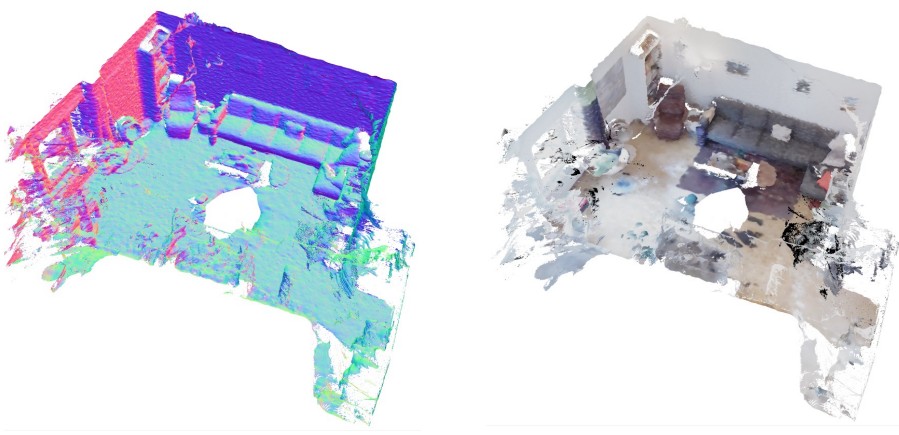

Figure 11: **Mesh reconstruction results on the self-collected scene without any optimization.**

