# OpenReview forum: "3D Reconstruction with Generalizable Neural Fields using Scene Priors"
_ICLR.cc/2024/Conference — ICLR 2024 poster_

### Official Review · Reviewer_zLQa · 2023-10-30

**Soundness:** 3 good
**Presentation:** 3 good
**Contribution:** 3 good
**Rating:** 6
**Confidence:** 4

**Summary:**

The paper proposes to learn a generalizable neural fields as a scene prior for 3D indoor scene reconstruction. The authors first sample a sparse point cloud from reprojected depth and use PointConv for extract per-point geometry feature which are then interpolated to a feature vector for any input point. Similarly, they use a CNN to extract texture features from the RGB image and splat the feature to the sampled 3D points. An MLP decoder predict the SDF and view dependent appearance from the point features. With the predicted SDF and appearance, they use volume rendering to render a depth/color for any input rays. The encoder and decoder are pretrained in a large scale dataset. After training, the encoder and decoder predict the SDF and appearance for new scenes without per-scene optimization. Of course, per-scene optimization improve the results as shown in the experiments.

**Strengths:**

1. The proposed method can do direct fusion by simply concatenate sampled point cloud from each input frame without additional fusion module thanks to the sparse point cloud representation of the scene.
2. The experimental results are extensive and show that the proposed method works well on both sparse input and dense input with faster convergence speed.

**Weaknesses:**

1. The model is not scale/rotation/translation invariant. I think the main reason is the use of point position as input to the decoder, which means if the coordinates system is changed, the output of the decoder is also changed. Similarly, the surface normal or view directions should also be in the local coordinates system, otherwise the output will change if the scene was translated. I wonder how sensitive of the current model to random rotation/translation. Would be great to have an ablation study. A simple solution is just don't use point position as input to the network. I am also interested in how this performs.

2. Would be great to cite and discuss [1] as it's very related to the paper. The paper already included many baselines so this is not a minus point.

[1] Fast Monocular Scene Reconstruction with Global-Sparse Local-Dense Grids

**Questions:**

1. One very simple baseline is missing: simply use TSDF fusion with the GT depth map for geometry reconstruction.
2. Why not also use CNN for the depth map and then use the similar method to get the geometry feature for the point cloud?
3. In Table 3. most of the baseline are not using depth as input, might be good to make it clear.
4. In the direct fusion part, points from different views are directly concatenated, will this change the distribution of point density and therefore change the K neighbouring points during inference?
5. Typo in the first paragraph "Although these results are encouraging, Although these results are encouraging".

---

> ### Author Response · Authors · 2023-11-20
> **Author Response to Reviewer zLQa (Part 1/2)**
>
> Thank you for your detailed comments. We address each of the concerns as follows
>
> ---
>
> **Q**: “The model is not scale/rotation/translation invariant. I think the main reason is the use of point position as input to the decoder, which means if the coordinates system is changed, the output of the decoder is also changed. Would be great to have an ablation study. A simple solution is just don't use point position as input to the network. I am also interested in how this performs.”
>
> **A**: Thanks for pointing out the drawbacks of using point positions during the first stage of training. The point position is used in the many per-scene optimization approaches[1,2,3], we keep it in our per-scene optimization stage as well. To ensure the decoders can be shared in both prior models and per-scene optimization, we have to keep the point position as inputs during the prior learning stage. However, we have made some efforts to mitigate the effect of scale/rotation/translation invariant. For example, we zero-center the input point cloud first and scale it into a unit sphere before feature extraction and positional encoding during both training and inference to ensure the scale and translation invariant. We don’t need to ensure the rotation is invariant, as the view-dependent features are required.
> If we ignore the per-scene optimization stage, the point position is not necessary.
>
> [1] Barron, Jonathan T., et al. "Mip-nerf: A multiscale representation for anti-aliasing neural radiance fields." Proceedings of the IEEE/CVF International Conference on Computer Vision. 2021.
>
> [2] Wang, Peng, et al. "Neus: Learning neural implicit surfaces by volume rendering for multi-view reconstruction." arXiv preprint arXiv:2106.10689 (2021).
>
> [3] Li, Zhaoshuo, et al. "Neuralangelo: High-Fidelity Neural Surface Reconstruction." Proceedings of the IEEE/CVF Conference on Computer Vision and Pattern Recognition. 2023.
>
> ---
>
> **Q**:”Would be great to cite and discuss [1] as it's very related to the paper. The paper already included many baselines so this is not a minus point.”
>
> **A**: Thanks for mentioning this relevant paper, we have cited it in the revised paper. Additionally, **we add a comparison with [1] in Table1**.
>
> [1] Fast Monocular Scene Reconstruction with Global-Sparse Local-Dense Grids
>
> ---
>
> **Q**: ”One very simple baseline is missing: simply use TSDF fusion with the GT depth map for geometry reconstruction.”
>
> **A**: In Table 2, we have shown the results of Bundle-Fusion, which is a better approach than TSDF fusion or RGB-D SLAM due to the advanced global pose alignment procedure. The better reconstruction results on 10 synthetic scenes demonstrate the superiority of our approach. In addition, we show the mesh reconstruction of a scene from ScanNet by employing the TSDF fusion under the same setting of ours-prior(the sparse view) in https://drive.google.com/file/d/1lJ54Zwo7OnJvdroCABtusKKYgjmGc-cq/view?usp=sharing. It can be observed that under the same setting of sparse views input, the TSDF fusion can only reconstruct a very coarse mesh even with the ground-truth depth.
>
> ---
>
> **Q**: “Why not also use CNN for the depth map and then use a similar method to get the geometry feature for the point cloud?”
>
> **A**: We use PointConv rather than CNN for the depth map for its better performance. As the depth map is inherently point clouds. To process points, PointConv is more effective than CNN for its (1) capability of processing unstructured data; (2) permulation invariance; (3) local geometry representation; and (4) flexibility in dealing with data of large variations of density or scale.
>
> ---
>
> **Q**: “In Table 3. most of the baseline are not using depth as input, might be good to make it clear.”
>
> **A**: We first clarify that Go-Surf in Table 3 is trained with ground-truth depth maps and we will emphasize this point in the main paper.
> Most of the methods [1,2,3] using depth maps focus on mesh reconstruction rather than novel view synthesis.
>
> [1] Yu, Zehao, et al. "Monosdf: Exploring monocular geometric cues for neural implicit surface reconstruction." Advances in neural information processing systems 35 (2022): 25018-25032.
>
> [2] Azinović, Dejan, et al. "Neural rgb-d surface reconstruction." Proceedings of the IEEE/CVF Conference on Computer Vision and Pattern Recognition. 2022.
>
> [3] Williams, Francis, et al. "Neural fields as learnable kernels for 3d reconstruction." Proceedings of the IEEE/CVF Conference on Computer Vision and Pattern Recognition. 2022.

---

> ### Author Response · Authors · 2023-11-20
> **Author Response to Reviewer zLQa (Part 2/2)**
>
> ---
>
> **Q**: “In the direct fusion part, points from different views are directly concatenated, will this change the distribution of point density and therefore change the K neighbouring points during inference?”
>
> **A**: During the direct fusion part, we simply collect all surface points and their feature from each view together. In other words, the operation of KNN and PointConv have already been employed on each view independently, thus the distribution of point density will not be unchanged.  We use this concatenated surface feature and positions to construct the surface implicit representation that could represent the entire scene instead of a single view as described in Section 3.2
>
> ---
>
> **Q**: “Typo in the first paragraph "Although these results are encouraging, Although these results are encouraging".”
>
> **A**: Thanks for pointing out this typo. We have fixed it in the revised paper.

---

> > ### Comment · Reviewer_zLQa · 2023-11-22
> >
> > Thank you so much for your response. I will keep my positive rating.
> >
> > I think it would be great to add an ablation study of the point position input.

---

### Official Review · Reviewer_ZqzA · 2023-10-31

**Soundness:** 3 good
**Presentation:** 1 poor
**Contribution:** 2 fair
**Rating:** 5
**Confidence:** 3

**Summary:**

This works proposes a scene reconstruction and novel-view synthesis method by learning scene priors that leverage ground-truth RGB-D data. The proposed novel method allows to efficiently integrate features from multiplve views in order to obtain an implicit neural representation of the scene's geometry and texture. This scene representation can further be used to render images from novel viewpoints. Experimental results on ScanNetV2 show how it outperforms many state-of-the-art scene reconstruction methods while using fewer images and less computation time, the latter thanks to a accurate initial estimate of the scene representation before the optimization step. For novel-view synthesis evaluated on real-world scenes, results shows this method is comparable or better than a number of well-known methods in the literature such as NeRF and IBRNet.

**Strengths:**

**Novelty and significance**
This work tackles two challenging tasks with one method: scene reconstruction and novel-view-synthesis. The authors propose a novel combination of techniques with clear advantage on some aspects compared to other state-of-the-art methods in each of the two tasks. It makes use of efficient representations (3D keepoints), and  is designed to work with any number of input views regardless of however many are used for training. It also leverages depth ground-truth very well in a two-stage pipeline. I believe the method itself is a solid contribution to the vision community.

**Soundness of method**

The presented methods are generally sound and the benefits of its design are clear. In general it makes a number of useful/practical design choices based on the types of indoor scenes it's applied to.

**Experimental results**

Experimental results on a real-world dataset, ScanNetV2, demonstrate strong results for scene reconstructions as well as novel-view synthesis in complex scenes. It achieves SoTA reconstruction performance while using fewer input images than a number of related methods, and similarly its novel-view synthesis performance is also better than strong baselines.
Finally, the ablation studies in the paper show that each of the learned priors is important to the overall performance.

**Weaknesses:**

**Presentation**

The paper needs to be substantially proof-read, as it is it's not ready for publication.


**Author claims**

The authors make a number of claims, namely:
- Per-scene optimization-based methods are "not scalable, inefficient, and unable to yield good results given limited views". I am not sure what they refer to by scalable, and also not sure evidence is presented to support the claim.
- Learning-based multi-view stereo methods, "their multi-view setting makes it less flexible to scale up and to broad applications". Again, I'm not sure evidence is shown that these methods do not scale up and are less broadly applicably. Perhaps precise pointers to the results or references would help.

**Experiments**
- Lack of expeirments with sparse views. Given that the method can technically reconstruct a scene it would be interesting to test it on these settings. It would be interesting to see the performance curve as a function of input views.
- Restricted to a single dataset (ScanNetV2). While it's certainly a good dataset to evaluate on, as it is based on real-world scenes, it would have been useful to see results on other recently used NVS datasets even if synthetic.
- Novel view synthesis experiments are evaluated on a small number of scenes only, with potentially high variance of results.

**Questions:**

**A number of questions and suggestions:**

- It's not clear what is the protocal for making a result bold in Table 1, making it diffucult to quickly see what performs best on each section of rows.

- How exactly is importance sampling used? I couldn't find a technical explanation for how it's used. In particular, how is it used for novel-view synthesis tasks? Is it only used during training of the networks? In other words, for evaluation, I understand rendering of a novel view is done by uniform sampling of the ray?

- The term generalizable is used extensively and I'm not sure what they refer to in many cases (e.g. generalizable features, generalizable representations and generalizable losses). I would kindly ask the authors to either reduce the use of the term or be a bit more precise when using it.

**Some missing references.**

On learning based novel-view synthesis.
- Trevithick, Alex, and Bo Yang. "Grf: Learning a general radiance field for 3d representation and rendering." Proceedings of the IEEE/CVF International Conference on Computer Vision. 2021.
- Yu, Alex, et al. "pixelnerf: Neural radiance fields from one or few images." Proceedings of the IEEE/CVF Conference on Computer Vision and Pattern Recognition. 2021.

On using GT depth for guiding training geometric and colour scene functions.
- Stelzner, Karl, Kristian Kersting, and Adam R. Kosiorek. "Decomposing 3d scenes into objects via unsupervised volume segmentation." arXiv preprint arXiv:2104.01148 (2021).

---

> ### Author Response · Authors · 2023-11-20
> **Author Response to Reviewer ZqzA (Part 1/2)**
>
> Thank you for your detailed comments. We address each of the concerns as follows
>
> ---
>
> **Q**: “Per-scene optimization-based methods are "not scalable, inefficient, and unable to yield good results given limited views". I am not sure what they refer to by scalable, and also not sure evidence is presented to support the claim.”
>
> **A**: Here, “scalable” refers to generalizing to large-scale data, e.g., thousands of scenes. Under this, per-scene optimization is NOT a scalable approach, since (1) it needs to train each individual scene independently, and (2) thousands of scenes means thousands of different networks. Thus, learning a single network shared by all the scenes (e.g., similar to ResNet, VIT, etc. applicable to any images), is the goal of NFPs, also see Fig.1 in the paper.
>
> ---
>
> **Q**: Learning-based multi-view stereo methods, "their multi-view setting makes it less flexible to scale up and to broad applications". Again, I'm not sure evidence is shown that these methods do not scale up and are less broadly applicably. Perhaps precise pointers to the results or references would help.
>
> **A**: Analogous to how many more RGB images are available on the Internet versus videos, we believe that using single-view RGB-D images will follow a similar pattern versus RGB-D videos. This will allow us to scale up our prior model more easily with diverse RGB-D images in the future as more such content becomes available.  We note that our current framework can be naturally extended to multi-view inputs by integrating all multiple views in the volume space, without requiring re-training of the encoder or decoder.
>
> For example, our NFPs can be trained using a collection of independent RGB-D images from random different scenes, they DO NOT have to be sequences, as against the muti-view stereos approaches. This brings significant benefits: (1). Analogous to how many more RGB images are available on the Internet versus videos, we believe that using single-view RGB-D images will follow a similar pattern versus RGB-D videos. (2). Individual RGB-D images will be much more divergent statistically than RGB-D videos, which contain redundancy between frames. This leads to better generalizability of our network. (3). Networks with single image input is much easier to implement and to train, compared to multi-view setting networks that require complex components, e.g., GRUs, to model the relations between frames.
>
> ---
>
> **Q**: “Lack of expeirments with sparse views. Given that the method can technically reconstruct a scene it would be interesting to test it on these settings. It would be interesting to see the performance curve as a function of input views.”
>
> **A**: We note we DO present the results on sparse view setting, which is evaluated on both the ScanNet and 10 synthetic data, in Table 2 and Table 5. The curve of mesh reconstruction with different frame sampling rates during training is shown in : https://drive.google.com/file/d/1zEH_j9e7Ca0HWw8fZvsYlM2PDVCRFvSD/view
>
> ---
>
> **Q**:”Restricted to a single dataset (ScanNetV2). While it's certainly a good dataset to evaluate on, as it is based on real-world scenes, it would have been useful to see results on other recently used NVS datasets even if synthetic.”
>
> **A**: Beyond ScanNet, we also evaluate our approach on 10 synthetic scenes proposed by NeuralRGBD as listed in Table 2 and also present a qualitative result of mesh reconstruction of a self-captured room as shown in Figure 10 in supplementary materials.
>
> ---
>
> **Q**: “Novel view synthesis experiments are evaluated on a small number of scenes only, with potentially high variance of results.”
>
> **A**: For a fair comparison, we follow the same setting of NerfingMVS and NeRFfusion using 8 scenes from ScanNet for the evaluation of novel view synthesis. Besides these 8 scenes, we also show some qualitative results of colored mesh of other Scannet scenes reconstructed via our approach in Figure 9. Additionally, we further show a reconstruction of a self-captured room in Figure 10.
>
> ---
>
> **Q**:”It's not clear what is the protocal for making a result bold in Table 1, making it diffucult to quickly see what performs best on each section of rows.”
>
> **A**: We highlight the best results column-wise, i.e., top-1 of each matric is in bold. Table 1 is split into three sections. In the first and second sections, we list the RGB- and RGB-D-based state-of-the-art, and the last section is our proposed method.

---

> ### Author Response · Authors · 2023-11-20
> **Author Response to Reviewer ZqzA (Part 2/2)**
>
> **Q** : How exactly is importance sampling used? I couldn't find a technical explanation for how it's used. In particular, how is it used for novel-view synthesis tasks? Is it only used during training of the networks? In other words, for evaluation, I understand rendering of a novel view is done by uniform sampling of the ray?
>
> **A**: We use the importance sampling proposed in NeuS, starting from 64 coarse samples (uniformly sampled from near to far), we iteratively add 16 samples each time based on weights computed with previously sampled points. We perform 3 steps of importance sampling during both training and inference. Additionally, we added 16 samples uniformly sampled from the region near to given ground-truth depth maps only during training.
>
> ---
>
> **Q**: “The term generalizable is used extensively and I'm not sure what they refer to in many cases (e.g. generalizable features, generalizable representations and generalizable losses). I would kindly ask the authors to either reduce the use of the term or be a bit more precise when using it.”
>
> **A**: Thanks for your suggestion, we will reduce some of “generalizable” and keep a consistent form for better understanding. Similar to Q1, 'generalizable' in this context means that a single learned network can be applied to numerous scenes. It also indicates that the features learned in the latent space effectively capture the common statistical characteristics of geometry and texture patterns prevalent in our visual world. Again, none of these can be obtained from per-scene optimization-based approaches.
>
> **Q**: Some missing references.
> **A**: Thanks for pointing out these missing references, we will add them to the related work section.

---

> ### Author Response · Authors · 2023-11-22
> **Following up on the post-rebuttal discussion**
>
> Dear Reviewer ZqzA,
>
> Thank you so much again for the detailed feedback. We’re approaching the end of the author-reviewer discussion period. However, there are no responses yet to our rebuttal.
>
> Please do not hesitate to let us know if there is any further information or clarification we can provide. We hope to deliver all the information in time before the deadline.
>
> Thank you!

---

> > ### Comment · Reviewer_ZqzA · 2023-11-22
> > **Thank you for your response.**
> >
> > I thank the authors for addressing my questions. Regarding,
> >
> > **Author claims**
> >
> > I appreciate the authors response elaborating on what you mean by scalable as well as generlizable. That said, these points are still unresolved in the paper itself which is what matters. It's important to be precise in the definitions and claims in the paper.
> > It is even more important that the work produces evidence for the claims. I understand conceptually what is meant by some methods not being scalable, however no clear evidence is provided of such limitations either by referencing other work or providing empirical results. An example of a result would be to show how (a) amortized (generalizable) methods such as IBRNet, etc. do not make good predictions beyond the number of views they are trained on, whereas your method does, and (b) insufficient trainig data (with enough views per scene) exists on to solve that issue.
> >
> > As for the term generalizable, I appreciate the clarification and I understand there are multiple related meanings to it. It would be beneficial to clarify them in the paper as well. I still also do not understand what is meant by a generalizable (geometric) loss.
> >
> > **Experimental results**
> >
> > After considering the discussion with other reviewers, I am further convinced this work would greatly benefit from additional datasets and, by extension, other related SoTA works. For instance, the DTU MVS Dataset is a common benchmark in this field. I also don't think the author's response fully addressed my concerns. Namely, I understand experiments with 30 to 40 views per scene (as shown in your tables) can be considered as sparse. However I meant even further sparse settings, such as 1 to 10 views.
> > If the authors provide results on DTU, as another reviewer suggested, they could directly kill two birds in one stone: additional the evaluated datasets with very sparse views and compare with results from published works, which would include very relevant pieces of work:
> > - MVSNeRF: Fast Generalizable Radiance Field Reconstruction from Multi-View Stereo (Chen et al. 2021)
> > - SinNeRF: Training Neural Radiance Fields on Complex Scenes from a Single Image (Xu et al. 2021)
> > - Depth-supervised NeRF: Fewer Views and Faster Training for Free (Deng et al. 2022).
> > - RegNeRF: Regularizing Neural Radiance Fields for View Synthesis from Sparse Inputs (Niemeyer et al. 2022).
> >
> > In summary I believe the biggest weaknesses of this work, and the reason my score remains the same are:
> > - Some key claims of this work (more scalable than other methods as it can be trained on single images) are not jusitified empirically.
> > - Insufficient experimental scope: in terms of data setting (very sparse views) and benchmarks (DTU), and missing recent SoTA methods.

---

> ### Author Response · Authors · 2023-11-22
> **Author Response to Reviewer ZqzA**
>
> We appreciate the feedback from the reviewer and address each of the concerns as follows.
>
> ---
>
> **About claims**
>
> We respectfully disagree that generalizable remains unresolved in our paper, or we didn’t provide evidence for this claim. It appears there is a misunderstanding regarding the scope of generalizability in our approach: instead of focusing on generalizability within a single scene (training on a set of frames to generalize to others within the same scene), the generalizability of our approach goes significantly BEYOND it. Our NFPs are designed to generalize across a broad range of indoor scenes. Once trained, our network can be inference on any unseen indoor scene, without the need for additional training. In other words, our NFPs can reconstruct a test scene video through feed-forward inference alone, even when all frames are previously unseen. Additionally, we **did compare with IBRNet and MVSNeRF** on the task of novel view synthesis in Table 3.
>
> For clarification, the inference-only capability of our network is thoroughly evaluated in Table 1, **without any optimization on a new scene** . These experimental results are strong evidence for the proof of our claim on generalizable. In contrast, most references cited by the reviewer, such as SinNeRF, Depth-supervised NeRF, and RegNeRF, focus on optimization specific to each scene. Their scope of generalizability, if present, is confined to individual scenes. Notably, none of these studies propose an inference-only network like ours.
>
> ---
>
> **About sparse view**
>
> We would like to emphasize that our demonstration of sparse view reconstruction occurs during the per-scene optimization stage. Utilizing fewer views is entirely feasible: in the most extreme cases, our NFPs require no views at all. Purely through inference, our network can achieve effective reconstruction. We note that evaluating sparse view can effectively demonstrate the generalizability within a single scene,  which is, however, not the scope of our approach, as aforementioned.
>
> Also, 40-50 views in Scannet videos represent roughly 10% of the total frames, similar to using 3-5 views in DTU scenes, which is also about 10% of their total frames.
>
> ---
>
> **DTU v.s ScanNet**
>
> As we claimed in our paper, our work focused on indoor scene reconstruction which is quite different from the object reconstruction as data provided by DTU. DTU is not applicable to our current NFPs network, which is trained on the Scannet training set. We will add the experiment to supp with a NFPs network trained on the DTU training set.
>
> We note that, however, the approaches proposed for DTU can hardly be applied to indoor scenes indirectly, as also suggested in [1,2]. This is because DTU, which focus on centered object, contains views that commonly share a large overlap region. In other words, the single object in DTU has been observed several times with the multi-view images. However, in the case of the indoor scene, there are lots of less observed regions than DTU as the video needs to cover the entire room. Furthermore, the indoor scene data contains lots of texture-less regions like floor, ceiling, and wall which makes the reconstruction way more challenging. To prove this point, we show an example of mesh reconstruction of scene0084 in ScanNet using the state-of-the-art approach SparseNeuS[3]. The result is shown here: https://drive.google.com/file/d/1CXZ9g1AGQGmQ0pjzH25AiN3u-LZhE5G5/view?usp=sharing
>
> Additionally, our approach can generalize well to novel scenes outside Scannet as demonstrated in Figure 10 in the supplementary materials.
>
> [1]Yu, Zehao, et al. "Monosdf: Exploring monocular geometric cues for neural implicit surface reconstruction." Advances in neural information processing systems 35 (2022): 25018-25032.
>
> [2] Guo, Haoyu, et al. "Neural 3d scene reconstruction with the manhattan-world assumption." Proceedings of the IEEE/CVF Conference on Computer Vision and Pattern Recognition. 2022.
>
> [3] Long, Xiaoxiao, et al. "Sparseneus: Fast generalizable neural surface reconstruction from sparse views." European Conference on Computer Vision. Cham: Springer Nature Switzerland, 2022.

---

> ### Author Response · Authors · 2023-11-23
> **Following up on the post-rebuttal discussion**
>
> Dear Reviewer ZqzA,
>
> As the discussion period is nearing its end, with only 2 hours remaining, we would greatly appreciate it if you could review our recent response. We believe that we have effectively addressed all of your previous concerns. We actively stand by for the last hours of the discussion phase. We hope to deliver all the information in time before the deadline.
>
> Thank you!

---

### Official Review · Reviewer_2i3p · 2023-11-01

**Soundness:** 2 fair
**Presentation:** 3 good
**Contribution:** 2 fair
**Rating:** 5
**Confidence:** 4

**Summary:**

This paper presents a generalizable 3D reconstruction framework from RGB-D sequences for indoor scenes. The key motivation is to design separate, progressive stages to learn the geometry field and color field. Experiments on various datasets have demonstrated the effectiveness of the design.

**Strengths:**

(1) Overall, the paper is well written and easy to follow.

(2) The paper demonstrates comprehensive experiments and compares with different state-of-the-art (SOTA) methods, to highlight the advantage of the proposed method.

**Weaknesses:**

(1) To me, the novelty of this paper is limited. The key design of the geometry prior module is similar to PointNeRF (also employs a distance-wise feature aggregation from 3D point clouds). Also, learning a geometric prior (SDF) then pruning to facilitate the texture field is applied in previous neural 3D reconstruction methods such as NeRFusion and SparseNeuS.

(2) The paper only conducts experiments on RGB-D sequences to demonstrate the generalizability. To me, the technical impact would be much higher if it also works well a RGB sequences, where obtaining precise geometry is challenging. For RGB-D sequences, some classic methods such as BundleFusion and COLMAP-MVS can achieve superior generalizability across different environments without any training.

(3) The advantage over Go-Surf is not convincing on both accuracy and speed.

**Questions:**

I would consider to improve my rating if the authors can address my concern especially on the novelty of this work presented in the weakness part.

---

> ### Author Response · Authors · 2023-11-20
> **Author Response to Reviewer 2i3p**
>
> Thank you for your detailed comments. We address each of the concerns as follows.
>
> ---
>
> **Q**: “To me, the novelty of this paper is limited. The key design of the geometry prior module is similar to PointNeRF (also employs a distance-wise feature aggregation from 3D point clouds). Also, learning a geometric prior (SDF) then pruning to facilitate the texture field is applied in previous neural 3D reconstruction methods such as NeRFusion and SparseNeuS.”
>
> **A**:  We thank the reviewer for pointing out the relevant references, however, we respectfully disagree on the limited novelty on top of them. Here, we highlight the key differences:
>
> First, Point-Nerf introduced a separately learned geomatric network to reconstruct the explicit point clouds. Once trained, the point cloud is fixed and detached from the succeeding learning process. This limits the performance as errors in the reconstructed point clouds are not directly optimizable through the gradient from the rendering loss or the per-scene optimization losses (they have resolved to point pruning and growing to fix it). Our approach fills the gap by joint learning of both geometry and texture representation, which is much simpler and more effective. Second, we note that both NeRFusion and SparseNeuS learned or optimized the geometry and texture field simultaneously. They either consider performing novel view synthesis(NeRFusion), where the geometry is not evaluated, or perform pure geometric reconstruction(SparseNeuS)
>
> Also, these three mentioned approaches are based on muti-view stereos, while our approach can be trained using a collection of independent RGB-D images from random different scenes, which DO NOT have to be sequences. This bring significant benefits: (1). Analogous to how many more RGB images are available on the Internet versus videos, we believe that using single-view RGB-D images will follow a similar pattern versus RGB-D videos. (2). Individual RGB-D images will be much more divergent statistically than RGB-D videos, which contains redundancy between frames. This leads to better generalizablily of our network. (3). Networks with single image input is much easier to implement and to train, compared to multi-view setting networks that require a complex component, e.g., GRUs, to model the relations between frames.
>
> ---
>
> **Q**: “The paper only conducts experiments on RGB-D sequences to demonstrate the generalizability. To me, the technical impact would be much higher if it also works well a RGB sequences, where obtaining precise geometry is challenging. For RGB-D sequences, some classic methods such as BundleFusion and COLMAP-MVS can achieve superior generalizability across different environments without any training.”
>
> **A**: While our approach is potentially applicable for monocular depth maps (e.g., aligned via camera pose), in this paper, we focus on how to fully utilize the depth prior. We will leave the extension to RGB sequences in future work.
> For inputting RGB-D sequences, first, it is NOT true that BundleFusion and COLMAP-MVS do not require training, rather, they do need to perform per-scene optimization for reconstruction. We note that unlike us, neither of them explored any generalizable geometric/texture representation. Our approach fills this gap.
> Secondly, as evidenced in Table 2, our approach outperforms both approaches with only 3% number of frames.
>
> ---
>
> **Q**: “The advantage over Go-Surf is not convincing on both accuracy and speed”
>
>
> **A**: We’d like to point out that the main contribution of our approach is to propose a generalizable scene prior that enables fast, large-scale scene reconstruction. As demonstrated in Table 1, without any optimization, our prior network can achieve comparable performance with some per-scene optimization approaches. Therefore, we emphasize our approach facilitates two things Go-Surf doesn’t provide. 1. We have a prior network that can reconstruct scenes by merely inferencing, no optimization is needed. 2. Single-view novel view synthesis as shown in Figure 7 in supplementary material (useful for image input on mobile devices)
>
> Also, our approach using per-scene optimization still outperforms Go-Surf in both aspects, specifically, the speedup is significant (e.g., use less than ½ of the time). And as demonstrated in Table 3 and Figure 4, Go-Surf focuses on geometric reconstruction, the NVS results are obviously less appealing.
>
> ---

---

> ### Author Response · Authors · 2023-11-22
> **Following up on the post-rebuttal discussion**
>
> Dear Reviewer 2i3p,
>
> Thank you so much again for the detailed feedback. We’re approaching the end of the author-reviewer discussion period. However, there are no responses yet to our rebuttal.
>
> Please do not hesitate to let us know if there is any further information or clarification we can provide. We hope to deliver all the information in time before the deadline.
>
> Thank you!

---

> ### Author Response · Authors · 2023-11-23
> **Following up on the post-rebuttal discussion**
>
> Dear Reviewer 2i3p,
>
> We apologize for reaching out again but feel compelled to express our concerns as the discussion period is nearing its end, with only 2 hours remaining.
>
> We would greatly appreciate it if you could review our recent response. We believe that we have effectively addressed all of your previous concerns. We actively stand by for the last hours of the discussion phase.  We hope to deliver all the information in time before the deadline.
>
> Thank you!

---

### Official Review · Reviewer_EoMe · 2023-11-01

**Soundness:** 3 good
**Presentation:** 3 good
**Contribution:** 3 good
**Rating:** 8
**Confidence:** 5

**Summary:**

The paper addresses high-fidelity 3D reconstruction using cross-dataset (ie, -scene in particular) generalization using the popular conjunction of neural radiance fields and signed distance functions.

The proposed method shows favorable comparative results on Scannet, a well established public benchmark in the field against RGB and RGBD based baselines.

**Strengths:**

+ ## Readability.
As it currently stands, the paper is very well written. The main ideas and concepts are mostly well explained and articulated throuthout.

+ ## Organization of the contents and overall paper structure.
The contents are also very well structured and balanced.

+ ## Related work section and discussion.
It is very well structued, articulated and populated with very relevant and up to date references.

+ ## The disclosed performance of the proposed method is at the very least competitive and promising.

+ ## The problem at hand (scene level generalization) is an important, impactfull one in the field.

**Weaknesses:**

+  ## 1. Missing bits of context information - How much does it cost?
While indicative timings and thorough implementation details (in supMat) are provided, information regarding the resource usage, model size and complexity are yet underdescribed.

A comparative disclosure of such information covering the main experimental baselines that are considered would help the reader better assess its relative positioning throughout the typical criteria.

Mentioning where the computation bottlenecks lie in terms of components would also be valuable in order to fully assess the practical usefullness of the proposed sequential pipeline, beyond rough timings (eg, Fig 3).

+  ## 2. Comparative evaluation - Baselines and Benchmarks.

While very recent work (eg, CVPR 2023) have been included in the setup, eg, HelixSurf (Liang et al.), there are a few missing players that currently hine by their absence.

For example, the RGB based references:

-- Li, Z., Müller, T., Evans, A., Taylor, R. H., Unberath, M., Liu, M. Y., & Lin, C. H. (2023). Neuralangelo: High-Fidelity Neural Surface Reconstruction. In Proceedings of the IEEE/CVF Conference on Computer Vision and Pattern Recognition (pp. 8456-8465).

-- Darmon, F., Bascle, B., Devaux, J. C., Monasse, P., & Aubry, M. (2022). Improving neural implicit surfaces geometry with patch warping. In Proceedings of the IEEE/CVF Conference on Computer Vision and Pattern Recognition (pp. 6260-6269).

-- Zhang, J., Yao, Y., Li, S., Fang, T., McKinnon, D., Tsin, Y., & Quan, L. (2022). Critical regularizations for neural surface reconstruction in the wild. In Proceedings of the IEEE/CVF Conference on Computer Vision and Pattern Recognition (pp. 6270-6279).

-- Wang, Y., Skorokhodov, I., & Wonka, P. (2022). Hf-neus: Improved surface reconstruction using high-frequency details. Advances in Neural Information Processing Systems, 35, 1966-1978.

Similarly, the DTU public benchmark could have been envisionend, albeit partially just to compare other key papers from the state-of-the-art without having to re-run their public implementations.

The same goes for the Tanks and Temples public benchmark as well. At least one additional benchmark would have been a reasonable addition.

This would help better - and more thoroughly - assess the relative positioning of the proposed contribution, performance-wise.

+  ## 3. The aforementioned references also lack in qualitative discussion and Related Work.

This is a direct consequence of (2) above.

**Questions:**

The main questions I would have cover the aforementioned weaknesses that have been pinpointed.

Besides those remaining grey areas, I would be happy to bump my initial rating were they to be addressed accordingly.

---

> ### Author Response · Authors · 2023-11-20
> **Author Response to Reviewer EoMe**
>
> Thank you for your detailed comments. We address each of the concerns as follows.
>
> ---
>
> **Q**: Missing bits of context information - How much does it cost?
>
> **A**: The geometric and texture priors network are trained on 8 NVIDIA V100 GPUs for 2 days until convergence. The per-scene optimization step is trained and tested on a single NVIDIA V100 GPU. All baselines reported in our paper are tested using the same computational resources.  As presented in Table 1&6 of our paper, during inference, the prior network reconstructs the entire scene within 5 minutes and the per-scene optimization network initialized by the prior network achieves sota performance within 15 minutes. Our approach yields superior computational efficiency thanks to the generalization representation.
>
> ---
>
> **Q**: Comparative evaluation - Baselines and Benchmarks.
>
> **A**: Thanks for mentioning these baselines, we will cite them in our revised paper. As many of them focus on reconstructing a texture object rather than an indoor scene, we try our best to adapt them on ScanNet and we find that most of them don’t work at all. This is because most of them have difficulty in handling low-textured or less-observed regions which are common in indoor scenes as suggested by [1,2]. Among these methods, only Neuralangelo[3] could work without major modifications. Beyond Neuralangelo, we further include another baseline[4]
>
> | Method | Training Time(mins) | Acc↓ | Comp↓ | Prec↑ | Recall↑ | F-score↑ |
> | --- | :---: | --- | --- | --- | --- | --- |
> | Neuralangelo | 150 | 0.084 | 0.071 | 0.611 | 0.605 | 0.609 |
> | FastMono | 30 | 0.042 | 0.056 | 0.751 | 0.678 | 0.710 |
> |Ours-prior | 5  | 0.084 | 0.057 | 0.695 | 0.764 | 0.737 |
> |Ours | 15 | 0.049 | 0.017 | 0.947 | 0.962 | 0.954 |
>
> [1]Yu, Zehao, et al. "Monosdf: Exploring monocular geometric cues for neural implicit surface reconstruction." Advances in neural information processing systems 35 (2022): 25018-25032.
>
> [2] Guo, Haoyu, et al. "Neural 3d scene reconstruction with the manhattan-world assumption." Proceedings of the IEEE/CVF Conference on Computer Vision and Pattern Recognition. 2022.
>
> [3] Li, Zhaoshuo, et al. "Neuralangelo: High-Fidelity Neural Surface Reconstruction." Proceedings of the IEEE/CVF Conference on Computer Vision and Pattern Recognition. 2023.
>
> [4] Dong, Wei, et al. "Fast Monocular Scene Reconstruction with Global-Sparse Local-Dense Grids." Proceedings of the IEEE/CVF Conference on Computer Vision and Pattern Recognition. 2023.

---

> ### Author Response · Authors · 2023-11-22
> **Following up on the post-rebuttal discussion**
>
> Dear Reviewer EoMe,
>
> Thank you so much again for the detailed feedback. We’re approaching the end of the author-reviewer discussion period. However, there are no responses yet to our rebuttal.
>
> Please do not hesitate to let us know if there is any further information or clarification we can provide. We hope to deliver all the information in time before the deadline.
>
> Thank you!

---

> ### Author Response · Authors · 2023-11-23
> **Following up on the post-rebuttal discussion**
>
> Dear Reviewer EoMe,
>
> We apologize for reaching out again but feel compelled to express our concerns as the discussion period is nearing its end, with only 2 hours remaining.
>
> We would greatly appreciate it if you could review our recent response. We believe that we have effectively addressed all of your previous concerns. We actively stand by for the last hours of the discussion phase.  We hope to deliver all the information in time before the deadline.
>
> Thank you!

---

### Official Review · Reviewer_s8AK · 2023-11-06

**Soundness:** 3 good
**Presentation:** 3 good
**Contribution:** 3 good
**Rating:** 6
**Confidence:** 4

**Summary:**

Input: One or more RGB-D images of an indoor scene

Output: Textured 3D mesh representing the indoor scene

The paper presents a generalizable neural framework, called Neural Field Priors (NFPs), for reconstructing 3D indoor scenes from a single as well as multiple RGB-D input images of the scene. Scene priors are obtained from depth map inputs, given posed RGB-D images. Results show significant performance improvement, especially on single-view 3D scene reconstruction, both in terms of speed and reconstruction quality.

The main contributions of the paper are two folds: (a) developing a two-stage generalizable neural framework using scene priors (i.e., not restricted to per-scene training) that is scalable to large-scale scenes, and (b) reconstructing the 3D scene by merging multiple view images (it is the features that are actually merged) in the volumetric space without using a fusion module.

The two-stage framework consists of a generalizable geometric prior and a generalizable texture prior.
The first network, which is the Geometric Prior network, is responsible for obtaining a signed distance field of the underlying scene. This is done by what are called Geometry Objectives and Surface Regularization. Geometry Objectives are based on the depth values. First, a pixel-wise rendering loss on depth maps is enforced to make the depth predictions at points sampled along a ray as close to the GT as possible (Importance sampling is used). The features of the points at the sampled locations are obtained using a modified form of weighted interpolation of surface-point geometric features. The surface points are nothing but the projection of depth image to 3D, and their features, a.k.a geometric features, are obtained by PointConv network. These interpolated point features, along with the point locations and their positional encoding, are passed through an MLP (called the Geometric decoder) to obtain signed distance values at the respective points. The signed distance value at a point is approximated to the GT SDF value by comparing the predicted SDF value with the difference of GT depth and predicted depth at that point. Surface Regularization is used for regularizing the SDF predictions to avoid artifacts. This is the Eikonal loss, which is a standard regularization term used in prior volume-rendering-based 3D reconstruction works.

The second network, which is the Texture Prior network, uses the SDF predictions from the first network as geometric initialization. The goal here is to learn RGB values for sampled points along the ray for which SDF values have been predicted by the Geometric Prior network. The texture features are a modified version of weighted interpolated surface-point texture features. The texture features here refer to convolutional features (image pixels to 3D correspondence yields surface-point convolutional features). These interpolated texture features for the points, along with the point locations and their positional encoding, are passed through an MLP (called the Texture decoder) to estimate the color at the respective points. During this process, the Geometric Decoder along with the PointConv encoder is jointly learned. So the loss for the Texture Prior network is the Geometric Prior loss plus the RGB loss.

When multi-view images are used as input, Geometric and Conv features of these images are merged during the reconstruction process. This avoids the burden of learning fusion modules to fuse reconstruction results from multiple views, thereby making the training efficient (less training complexity). As well, it is claimed that it allows for flexible data processing (I have a few questions on this in the Questions section).

Dataset used:
ScanNet_v2 and 10 synthetic scenes from Azinovic et al. 2022

Underlying Neural Network:
Geometry encoder: PointConv
Image Encoder: U-Net
Decoders are MLPs
Volume rendering is what makes this possible

Loss function:
L_depth (L1 loss), L_sdf (L1 loss), L_surface (Eikonal loss), L_rgb (L2 loss)

Quantitative Metric:
3D scene mesh reconstruction – Accuracy, Completeness, Precision, Recall, F-score
Novel view synthesis – PSNR (Power Signal-to-Noise ratio), SSIM (Structural Similarity for Image comparison metric) and LPIPS (Learned Perceptual Image Patch Similarity metric)

**Strengths:**

1)	Generalized framework for scene reconstruction using radiance fields (i.e., no per=scene training) from relatively few input views (relative to existing literature)

2)	Ability to reconstruct a 3D scene by merging individual frames in the volumetric space without a learnable fusion module

3)	Novel view synthesis from single-view input beats existing works

4)	Simple interpolation strategy for obtaining point features. Making use of surface points instead of dense volumetric grids for obtaining sampled point features

5)	The paper is well-written

**Weaknesses:**

1)	The dependency on depth maps is limiting since such data is not always available. This is also a drawback of MonoSDF and other works that use additional priors for scene reconstruction, beyond just RGB images
2)	The intuition behind the approximation of GT SDF values by observing depth values along a ray is unclear. This may result in erroneous signed distance predictions. An explanation of this is lacking. I am actually interested in this ablation experiment
3)	Results on images in the wild are missing. This will add value to the work
4)	Limited quantitative results
5)	Ablation experiments are not thorough, in terms of the different components involved in Geometric and Texture Prior Networks
6)	Discussion on limitations is lacking

**Questions:**

1)	It is claimed in the second+third paragraph of the Introduction that not having a fusion module to handle multi-view images during training allows for flexible data processing. This is not substantiated and remains unclear to me. Can you elaborate how this is the case?
2)	Is there a reason behind using PointConv as geometric feature embedding network? Were alternative networks (like PointNet, DGCNN etc.) tried? Using different networks should have a bearing on the overall result. Were any experiments conducted to understand this?
3) This has been touched upon but I would like to ask again -- what makes the proposed approach work on single images? To what extent are the reconstructed scenes reasonable? Or put differently, what "gaps" needs to be "filled in" to make the reconstruction results (from single input image) better than what can be currently achieved?

---

> ### Author Response · Authors · 2023-11-20
> **Author Response to Reviewer s8AK**
>
> Thank you for your detailed comments. We address each of the concerns as follows.
>
> ---
>
> **Q**: “The dependency on depth maps is limiting since such data is not always available. This is also a drawback of MonoSDF and other works that use additional priors for scene reconstruction, beyond just RGB images”
>
> **A**: We acknowledge that depth data is not universally available, however, it's important to note that RGBD settings are increasingly common, e.g., in applications in robotics, self-driving cars, augmented reality, etc., where depth sensors are often integrated. Our method uses depth data to create compact surface representations, rather than only as objectives like MonoNerf, offering a more efficient alternative to the usual volumetric approaches.
>
> ---
>
> **Q**: “The intuition behind the approximation of GT SDF values by observing depth values along a ray is unclear. This may result in erroneous signed distance predictions. An explanation of this is lacking. I am actually interested in this ablation experiment”
>
> **A**: Using depth maps to approximate the SDF value has already been applied in previous work, e.g., [1,2], based on the fact that when only depth is provided, computing the exact distance of a location to the surface is infeasible. To resolve it, we have to compute the distance to a single, nearby surface point, which yields an approximated value that is equal or larger than the ground truth, i.e., an up-bound for the SDF value. Thus, to establish a tight bound, we choose to use the distance along the ray between the query point depth and the measured depth, as described in our paper.
>
> [1] Ortiz, Joseph, et al. "isdf: Real-time neural signed distance fields for robot perception." arXiv preprint arXiv:2204.02296 (2022).
> [2] Azinović, Dejan, et al. "Neural rgb-d surface reconstruction." Proceedings of the IEEE/CVF Conference on Computer Vision and Pattern Recognition. 2022.
>
> ---
>
> **Q**: “Results on images in the wild are missing. This will add value to the work”
>
> **A**: Figure 10 in the supplementary material shows the reconstruction results of a self-captured room. It can be considered as the images in the wild
>
> ---
>
> *Q**:Limited quantitative results
>
> **A**: In the main paper and supplementary, we show several quantitative results on real-world and synthetic data, including comparisons for mesh reconstruction on ScanNet(Table1), comparisons for mesh reconstruction on 10 synthetic data(Table2), comparisons for novel view synthesis on ScanNet(Table3), mesh reconstruction with sparse view(Table5), comparisons with other generalizable method for mesh reconstruction(Table7). In addition, we show another ablation study in the following section.
>
> ---
>
> **Q**: Ablation experiments are not thorough, in terms of the different components involved in Geometric and Texture Prior Networks
>
> **A**: In Sec. 4.3, we did show the ablation studies to evaluate the impact of Geometric and Texture prior networks, providing essential validation for their functionality. We note that loss functions such as $L_{eik}$ is commonly used and thoroughly analyzed in the previous literature[1,2,3]. Removing any loss in training the Geometric prior, such as $L_{depth}$ or $L_{sdf}$ will lead to inconvergence of the network; We conduct an ablation study w./w.o. the refined surface feature as described in the last paragraph of Page 4. The comparisons are listed below,
>
> | Method | Acc↓ | Comp↓ | Prec↑ | Recall↑ | F-score↑ |
> | --- | --- | --- | --- | --- | --- |
> | w./o. refine mlp | 0.089 | 0.062 | 0.675  | 0.728 | 0.707 |
> | w. Refine mlp | 0.084 | 0.057 | 0.695 | 0.765 | 0.737 |
>
> [1]Yu, Zehao, et al. "Monosdf: Exploring monocular geometric cues for neural implicit surface reconstruction." Advances in neural information processing systems 35 (2022): 25018-25032.
>
> [2] Guo, Haoyu, et al. "Neural 3d scene reconstruction with the manhattan-world assumption." Proceedings of the IEEE/CVF Conference on Computer Vision and Pattern Recognition. 2022.
>
> [3] Wang, Peng, et al. "Neus: Learning neural implicit surfaces by volume rendering for multi-view reconstruction." arXiv preprint arXiv:2106.10689 (2021).
>
> ---
>
> **Q**:Discussion on limitations is lacking
>
> **A**: The performance of the prior network could be further improved. Currently, we only utilize very shallow encoders, such as PointConv and ResNet50. Some advanced architectures can be utilized as well. Additionally, our prior models are only trained on ScanNetV2 with 1213 scenes. As mentioned in the paper, since the network takes in single images instead of videos, it is easy to scale up the training data. We expect the performance could be further improved with the upgrade of the network architecture and the increase of training data
>
> ---

---

> > ### Comment · Reviewer_s8AK · 2023-11-20
> > **Rebuttal Acknowledgment**
> >
> > Dear Authors,
> >
> > Thank you for the point-wise answers to my questions.
> > My concerns regarding Geometric/Texture prior networks' ablations and approximating GT SDF values with depth values along a ray are addressed.
> >
> > I will keep my positive score for the paper.

---

> > > ### Author Response · Authors · 2023-11-21
> > > **Thank you!**
> > >
> > > Dear Reviewer s8AK,
> > >
> > > Thank you so much for your positive feedback. We appreciate the time and effort you have put into reviewing our modifications and responses. And we are glad that the questions have been resolved!

---

### Author Response · Authors · 2023-11-20
**General Response**

Dear reviewers, we appreciate all the detailed comments and helpful suggestions. We have highlighted the changes in red in the revised version of our paper. Specifically, we have made the following updates:

- We have added the ablation of a component used in geometric prior learning to address the concerns of Reviewer s8AK.
- We have added FastMono as another baseline in Table 1, following Reviewer EoMe and Reviewer zLQa’s comments.
- We have discussed the computational resource used for training and inference in Appendix B.1 following Reviewer EoMe’s suggestions
- We have clarified the process of importance sampling as suggested by Reviewer ZqzA in Appendix A.1

---

**About single-view training setting**

Analogous to how many more RGB images are available on the Internet versus videos, we believe that using single-view RGB-D images will follow a similar pattern versus RGB-D videos. This will allow us to scale up our prior model more easily with diverse RGB-D images in the future as more such content becomes available. We note that our current framework can be naturally extended to multi-view inputs by integrating all multiple views in the volume space, without requiring re-training of the encoder or decoder.

For example, our NFPs can be trained using a collection of independent RGB-D images from random different scenes, they DO NOT have to be sequences, as against the muti-view stereos approaches[1,2,3]. This brings significant benefits: (1). Analogous to how many more RGB images are available on the Internet versus videos, we believe that using single-view RGB-D images will follow a similar pattern versus RGB-D videos. (2). Individual RGB-D images will be much more divergent statistically than RGB-D videos, which contain redundancy between frames. This leads to better generalizability of our network. (3). Networks with single image input are much easier to implement and train, compared to multi-view setting networks that require complex components, e.g., GRUs, to model the relations between frames.

[1] Xu, Qiangeng, et al. "Point-nerf: Point-based neural radiance fields." Proceedings of the IEEE/CVF Conference on Computer Vision and Pattern Recognition. 2022.

[2] Zhang, Xiaoshuai, et al. "Nerfusion: Fusing radiance fields for large-scale scene reconstruction." Proceedings of the IEEE/CVF Conference on Computer Vision and Pattern Recognition. 2022.

[3] Long, Xiaoxiao, et al. "Sparseneus: Fast generalizable neural surface reconstruction from sparse views." European Conference on Computer Vision. Cham: Springer Nature Switzerland, 2022.

---

### Meta-Review · Area_Chair_NTrX · 2023-12-07

**Metareview:**

- Claims and findings:

This submission introduces an approach to learn a generalizable neural field as a scene prior. Borrowing from previously existing approaches like PointNerf, NeRFusion and SparseNeuS. The authors seemed to partially overclaim the merits of their approach in the initial submission but has been updated in the rebuttal. The submission includes comprehensive experiments and compares with different state-of-the-art (SOTA) methods achieving successful performance.


- Strengths:
Reviewers have highlighted that the paper is well written and easy to follow. In addition, reviewers have also noted work tackles two challenging tasks with one method: scene reconstruction and novel-view-synthesis and that the method itself is a solid contribution to the vision community.

- Weaknesses:
The paper only conducts experiments on RGB-D sequences to demonstrate the generalizability. Reviewers have pointed out that the technical impact would be much higher if it also works well a RGB sequences.


- Missing in submission:
Reviewer 2i3p pointed out that having access to GT depth might not realistic in some settings. I agree, it would be really interesting if the authors could have provided a version of their method using predicted monocular depth from a pre-trained model. In addition, reviewers note that benchmarking on the 7 scenes dataset would provide more insights as to the quality of the contribution.

**Justification For Why Not Higher Score:**

The submission is borderline and some reviewers have expressed concerns about missing benchmarks (see weaknesses and missing in submission).

**Justification For Why Not Lower Score:**

Reviewer scores are borderline leaning towards positive. I believe the concerns expressed by reviewer ZqzA have been partially addressed by authors.

---

### Decision · Program_Chairs · 2024-01-16

Accept (poster)